# High performance tandem organic solar cells via a strongly infrared-absorbing narrow bandgap acceptor

Zhenrong Jia[1,2,8], Shucheng Qin[1,2,8], Lei Meng [1,2✉], Qing Ma[1,2], Indunil Angunawela[3], Jinyuan Zhang[1], Xiaojun Li[1,2], Yakun He[4,5], Wenbin Lai[1,2], Ning Li [4,6], Harald Ade [3✉], Christoph J. Brabec [4,6] & Yongfang Li [1,2,7✉]

Tandem organic solar cells are based on the device structure monolithically connecting two solar cells to broaden overall absorption spectrum and utilize the photon energy more efficiently. Herein, we demonstrate a simple strategy of inserting a double bond between the central core and end groups of the small molecule acceptor Y6 to extend its conjugation length and absorption range. As a result, a new narrow bandgap acceptor BTPV-4F was synthesized with an optical bandgap of 1.21 eV. The single-junction devices based on BTPV-4F as acceptor achieved a power conversion efficiency of over 13.4% with a high short-circuit current density of 28.9 mA cm$^{-2}$. With adopting BTPV-4F as the rear cell acceptor material, the resulting tandem devices reached a high power conversion efficiency of over 16.4% with good photostability. The results indicate that BTPV-4F is an efficient infrared-absorbing narrow bandgap acceptor and has great potential to be applied into tandem organic solar cells.

[1] Beijing National Laboratory for Molecular Sciences, CAS Key Laboratory of Organic Solids, Institute of Chemistry, Chinese Academy of Sciences, 100190 Beijing, China. [2] School of Chemical Science, University of Chinese Academy of Sciences, 100049 Beijing, China. [3] Department of Physics and Organic and Carbon Electronics Laboratories (ORaCEL), North Carolina State University, Raleigh, NC, USA. [4] Institute of Materials for Electronics and Energy Technology (i-MEET), Department of Materials Science and Engineering, Friedrich-Alexander University Erlangen-Nürnberg, 91058 Erlangen, Germany. [5] Erlangen Graduate School in Advanced Optical Technologies (SAOT), Paul-Gordan-Straße 6, 91052 Erlangen, Germany. [6] Helmholtz-Institute Erlangen-Nürnberg for Renewable Energy (HI ERN), Immerwahrstr. 2, 91058 Erlangen, Germany. [7] Laboratory of Advanced Optoelectronic Materials, College of Chemistry, Chemical Engineering and Materials Science, Soochow University, 215123 Suzhou, Jiangsu, China. [8] These authors contributed equally: Zhenrong Jia, Shucheng Qin. ✉email: menglei@iccas.ac.cn; hwade@ncsu.edu; liyf@iccas.ac.cn

Organic solar cells (OSCs) have drawn much attention in the past decade for its potential application as a reliable clean energy source[1–5]. Due to its advantages of light weight, flexibility, and low-cost solution processability, OSCs could realize commercialization in the near future[6–8]. "Tandem" architecture has been widely adopted as a simple and reliable methodology to achieve higher performing OSCs with better utilization of near-infrared (NIR) solar energy[9–13]. For a single-junction OSC with a specific absorption region and bandgap, the trade-off between thermalization loss of photon energy and spectrum unitization is limited on a certain level[14–16]. While in the series-connected tandem OSCs, the absorption spectrum wavelength region can be effectively extended by employing a wide bandgap sub-cell to harvest high energy photons and another narrow bandgap sub-cell for utilizing low energy photons. At the same time, the open-circuit voltage ($V_{oc}$) of the tandem OSCs is the summation of those of the two sub-cells[17–19]. And the thickness of each sub-layers can be easily tuned separately to match a balanced absorbance in each wavelength region and provide maximized photon-to-energy efficiency[13,20,21].

In 2013, Yang et al. designed the narrow bandgap polymer donor PDTP-DFBT with NIR absorption through molecular modification. PDTP-DFBT:PC_{71}BM was then adopted as the rear cell active layer and P3HT:ICBA as the front cell active layer to fabricate tandem OSC, leading to an NREL certified power conversion efficiency (PCE) of 10.6%, which was the first time of the PCE of the OSCs exceeding 10%[22]. Later on, in order to broaden the absorption, Hou et al. synthesized IEICO-4F acceptor with fluorine substituted end groups[23]. Combined with J52-2F:IT-M as front cell, the tandem OSC reached PCE of 14.9%[24]. By introducing Cl atoms into small molecule acceptor based on BDT units, Forrest et al. synthesized BT-CIC acceptor with absorption range over 900 nm and the PCE of tandem OSCs reached 15%[25,26]. Recently, Chen et al. adopted PTB7-Th:O6T-4F:PC_{71}BM ternary system as a rear cell material with absorption

spectra over 1000 nm and a high PCE of 17.3% was reported for the tandem OSCs[27].

At the current stage, the PCE of tandem OSCs still holds no advantage over single-junction OSCs[28]. One main limitation for the development of tandem OSCs comes from the lack of efficient low bandgap materials for rear cells, which mostly determines the effectiveness of utilizing the NIR region of the solar spectrum[29–32]. Based on previously reported theoretical simulations, the most suitable active layer materials for rear cells should possess an NIR absorption edge around 1050–1150 nm[33]. However, absorption onset of most reported narrow bandgap acceptors could not surpass 1000 nm without losing substantial external quantum efficiencies (EQEs) response in the NIR region[34–36]. Therefore, to meet the requirements raised by tandem OSCs, more effort needs to be devoted to developing ultra-narrow bandgap materials that can simultaneously offer a suitable absorption spectrum with high short-circuit current densities ($J_{sc}$) and low energy loss.

In recent years, the A-D-A structured narrow bandgap small molecule acceptors with absorption edge at ca. 800 nm have attracted great attention for their application in OSCs[37–40]. The A-D-A molecules are composed of a fused ring central donor (D) unit and two strong electron-accepting (A) end groups with an ethylene double bond π-bridges between them. Our group extended the absorption edge of the A-D-A structured small molecule acceptors to 850–900 nm by inserting another ethylene double bond π-bridges between the central fused ring D unit and the A end groups[41,42], indicating that using two enthylene double bonds π-bridge is an effective way to further redshift the absorption of the A-D-A acceptors. By inserting an electron-accepting (A′) core in the central fused ring D unit, Zou et al. recently synthesized an A-DA′D-A small molecule acceptor Y6 with absorption edge extending to ca. 950 nm[41]. And the OSCs with Y6 as acceptor and wide bandgap polymer donor demonstrated high PCE of over 16%[43–45]. Based on the above consideration, in this work, we designed and synthesized a new low

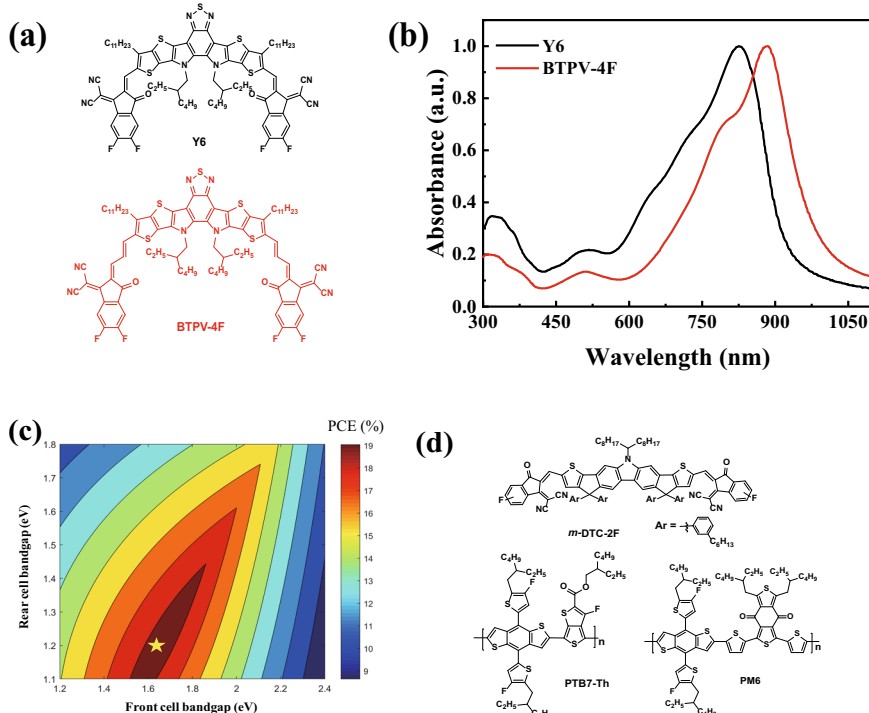

**Fig. 1 Materials design and characterization. a** Molecular structures of the acceptors BTPV-4F and Y6. **b** Absorption spectra of the BTPV-4F and Y6 films. **c** Simulated optimal bandgap matching of two sub-cells in tandem OSCs. **d** Molecular structures of the acceptor *m*-DTC-2F and the polymer donors PTB7-Th and PM6.

bandgap acceptor BTPV-4F (see Fig. 1a) by inserting another ethylene double bond π-bridge between the DA'D central fused ring unit and the A end groups of Y6 for further redshifting its absorption spectrum. BTPV-4F shows a significantly redshifted absorption spectrum covering from 600 nm to 1050 nm. The optical bandgap of BTPV-4F was determined to be 1.21 eV based on the onset wavelength (1021 nm) of the UV–Vis absorption spectrum of BTPV-4F film. The OSCs based on BTPV-4F as acceptor demonstrated a PCE of 13.4% with a high $J_{sc}$ of 28.9 mA cm$^{-2}$ due to the strong and wide NIR absorption of BTPV-4F acceptor. 28.9 mA cm$^{-2}$ is so far the highest $J_{sc}$ in the entire field of OSCs.

For the purpose of building highly efficient tandem OSCs, a wide bandgap sub-cell with suitable absorption range and low $V_{oc}$ loss is also critical. Through the simulation of optical bandgaps of rear and front cells, a front cell with a wide bandgap of ca. 1.6 eV should be selected to ideally match the rear cell based on BTPV-4F. In this case, a new acceptor m-DTC-2F was developed with bandgap of 1.61 eV. The single-junction OSCs based on m-DTC-2F as the acceptor demonstrated a PCE of 12.2% with a high $V_{oc}$ of 1.00 V and a $J_{sc}$ of 17.1 mA cm$^{-2}$. Eventually, a high PCE of 16.40% was achieved for the tandem OSCs.

## Results

**Material design and characterization.** Figure 2 shows the synthetic route of BTPV-4F and its detailed synthetic processes and characterization are described in the "Methods" section. To study the influence of inserting one more double bond π-bridge on optical and electrical properties of the acceptor, we performed theoretical calculation and frontier orbitals simulations by using density functional theory. The calculation results show that the insertion of two double bonds can effectively extend the conjugation length and form a larger π conjugated system (Supplementary Fig. 1). The calculated lowest unoccupied molecular orbital (LUMO) and the highest occupied molecular orbital (HOMO) electronic density distributions of BTPV-4F and Y6 were shown in Supplementary Fig. 2, which reveals the obviously narrower bandgap of BTPV-4F. In the meanwhile, Simulated molecular geometries of BTPV-4F and Y6 show almost similar planarity, which indicates that the introduction of one more double bond could largely maintain the conformation of the original Y6 acceptor molecules (Supplementary Fig. 3). Supplementary Fig. 4a shows the absorption spectrum of BTPV-4F in chloroform solution. The absorption spectrum of BTPV-4F film, as shown in Fig. 1b, redshifted for more than 100 nm than its solution (Supplementary Fig. 4a), indicating strong intermolecular interaction existed in BTPV-4F film. The absorption spectrum of BTPV-4F film also has a significant redshift in comparison with Y6 film (Fig. 1b): its absorption peak is redshifted from 821 nm to 887 nm and absorption edge is redshifted to 1050 nm corresponding to an $E_g^{opt}$ of 1.21 eV. Furthermore, The extinction coefficients of BTPV-4F and Y6 in chloroform solution are $1.52 \times 10^5$ L mol$^{-1}$ cm$^{-1}$ and $1.31 \times 10^5$ L mol$^{-1}$ cm$^{-1}$, respectively. The extinction coefficients of BTPV-4F and Y6 films are $1.21 \times 10^5$ cm$^{-1}$ and $1.09 \times 10^5$ cm$^{-1}$, respectively. BTPV-4F shows higher extinction coefficients in both solution and film than Y6 (Supplementary Fig. 4).

For the eventual purpose of fabricating high-performance tandem OSCs, another important issue is to find the wide bandgap photovoltaic materials for the front cell with complementary absorption with the rear cell based on BTPV-4F. Considering the fact that the PCE of tandem OSCs is limited by $J_{sc}$ within a certain range of solar spectrum, an optical simulation was performed to guide the determination of the optimal bandgap matching for front cells and rear cells[35]. As shown in Fig. 1c, a simulated high PCE over 19% could be possibly achieved (with assumed $V_{oc}$ loss of 0.55 V, average EQE of 75% and FF of 75%), and the expected most suitable bandgap for front cell materials should be around 1.6 eV in matching with the bandgap of ca. 1.2 eV for the rear cell, based on the simulation. Recently, Hsu et al. reported a carbazole based acceptor with two fluorine on the IC end groups called DTC (4Ph)-4FIC[40]. The PM6: DTC(4Ph)-4FIC device shows high $V_{oc}$ of 0.95 V and over 800 nm EQE response. In order to simultaneously get ideal bandgap and higher $V_{oc}$, we accordingly designed and synthesized a carbazole based small molecule acceptor m-DTC-2F with single fluoro-substitution IC end groups and *meta*-alkyl-phenyl side chains (Fig. 1d). Figure 3 shows the synthetic route of m-DTC-2F and the detailed synthetic processes and characterizations of *m*-DTC-2F can be found in the "Methods" section. The optical bandgap of m-DTC-2F is determined to be 1.61 eV from its absorption edge as shown in the absorption spectrum of m-DTC-2F film in Supplementary Fig. 5, which would be able to ideally match the bandgap (1.21 eV) of BTPV-4F in the rear cell.

Cyclic voltammetry measurements were performed to estimate the electronic energy levels of the two sub-cells, as shown in Supplementary Fig. 6. Physicochemical properties and electronic energy levels of acceptors are listed in Supplementary Table 1. From the onset oxidation potential ($E_{ox}$) and onset reduction potential ($E_{red}$), HOMO energy level ($E_{HOMO}$) and the LUMO energy level ($E_{LUMO}$) were determined according to the equation $E_{LUMO/HOMO} = -e \ (E_{red/ox} + 4.36)$ (eV) where the unit of $E_{red/ox}$ is V vs. Ag/AgCl. (Redox potential of Fc/Fc$^+$ is 0.44 V vs. Ag/AgCl in our measurement system (see Supplementary Fig. 6), and we take the energy level of Fc/Fc$^+$ as 4.8 eV below vacuum.) The $E_{LUMO}$ and $E_{HOMO}$ of m-DTC-2F were estimated to be −3.89 and −5.67 eV, and those values of BTPV-4F were −4.08 and −5.39 eV, respectively. Both cases in DFT calculation and cyclic voltammetry measurements, BTPV-4F shows relatively higher $E_{HOMO}$, similar $E_{LUMO}$ and narrower bandgap than Y6. Such a trend indicates that the insertion of double bonds can effectively upshift the HOMO level and reduce the bandgap of acceptors. In considering the complementary absorption and well-matched energy levels of the donor and acceptor materials (see Supplementary Fig. 6), the wide bandgap polymer PM6 was selected as the polymer donor for the front cell with m-DTC-2F as acceptor and the narrow bandgap polymer PTB7-Th was used as the polymer donor for the rear cell with BTPV-4F as acceptor.

**Fig. 2 BTPV-4F synthesis.** The synthetic route of BTPV-4F.

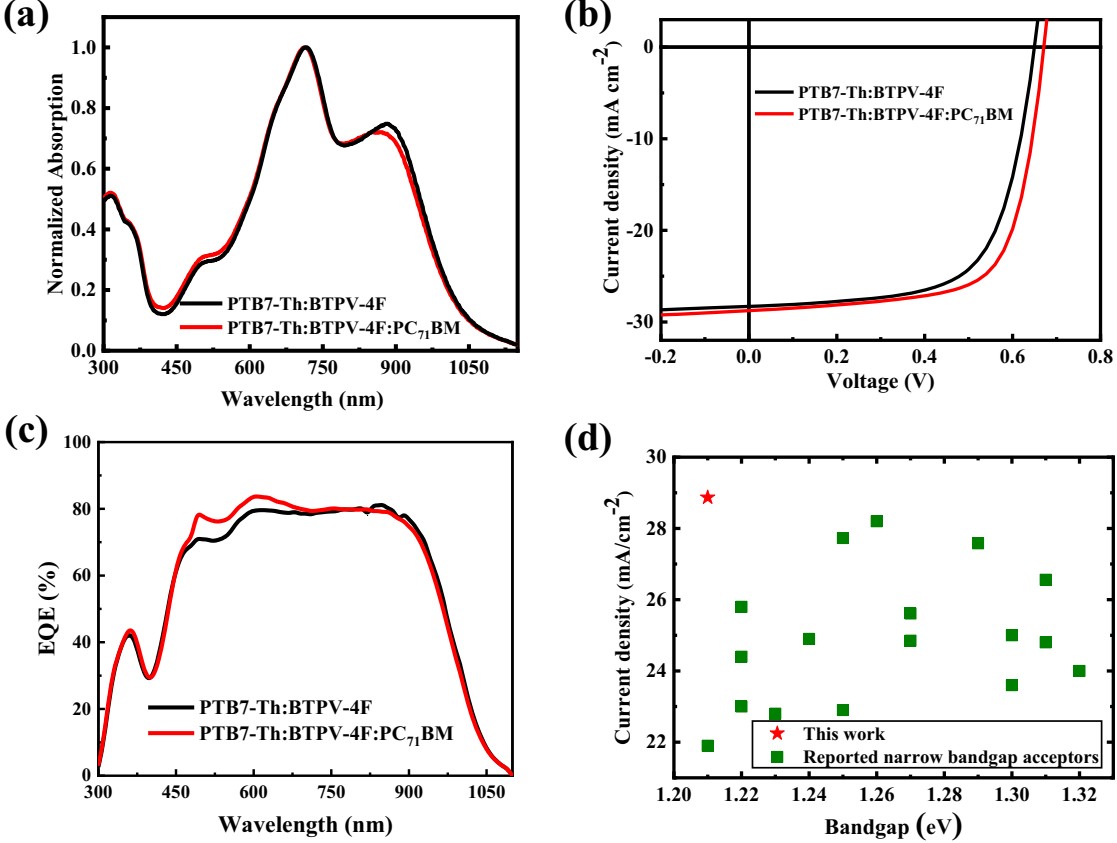

**Fig. 3 m-DTC-2F synthesis.** Synthetic route of m-DTC-2F.

**Fig. 4 Absorption spectra and photovoltaic performance of PTB7-Th:BTPV-4F. a** Normalized absorption spectra of blend films of PTB7-Th:BTPV-4F (1:1.5, w/w) and PTB7-Th:BTPV-4F:PC$_{71}$BM (1:1.5:0.15, w/w/w). **b** The J–V curves of the optimized binary OSC based on PTB7-Th:BTPV-4F(1:1.5, w/w) and ternary OSC based on PTB7-Th:BTPV-4F:PC$_{71}$BM (1:1.5:0.15, w/w/w). **c** The EQE spectra of the corresponding binary and ternary OSCs. **d** Plots of J$_{sc}$ values of the OSCs vs. optical bandgaps of the narrow bandgap acceptors used in the OSCs reported in the literature and from this work with the acceptor of BTPV-4F.

**Photovoltaic properties of narrow bandgap acceptor BTPV-4F.** To study the photovoltaic performance of the narrow bandgap acceptor BTPV-4F, we fabricated the inverted structure OSCs with BTPV-4F as acceptor and PTB7-Th as polymer donor[46] with the device structure of ITO/ZnO/active layer/MoO$_3$/Ag. In order to further improve the photovoltaic performance, we also fabricated ternary OSCs with PC$_{71}$BM as the second acceptor (or the third component)[47]. The optimized weight ratios of the photovoltaic materials in the active layers are 1:1.5 for the binary PTB7-Th: BTPV-4F active layer and 1:1.5:0.15 for the ternary PTB7-Th: BTPV-4F:PC$_{71}$BM active layer. Figure 4a shows the absorption spectra of the blend films of PTB7-Th:BTPV-4F (1:1.5) and PTB7-Th:BTPV-4F:PC$_{71}$BM (1:1.5:0.15). The blend films exhibit strong absorption in the NIR region. The fabrication conditions of the OSCs were optimized and the results are listed in Supplementary Tables 2–6. Figure 4b shows the current density–voltage (J–V) curves of the optimized devices under the illumination of AM1.5 G, 100 mW cm$^{-2}$, and the detailed photovoltaic parameters are listed in Table 1. The binary OSC based on PTB7-Th:BTPV-4F shows a PCE of 12.1% with a V$_{oc}$ of 0.65 V, a high J$_{sc}$ of 28.3 mA cm$^{-2}$, and an FF of 65.9%. After introducing PC$_{71}$BM as the third component, the best PCE of the ternary OSC based on PTB7-Th:BTPV-4F: PC$_{71}$BM with a weight ratio of 1:1.5:0.15 reached 13.4% with simultaneously enhanced V$_{oc}$ of 0.67 V, J$_{sc}$ of 28.9 mA cm$^{-2}$, and FF of 69.3%, compared to those of the binary device. The enhanced V$_{oc}$ could be due to the reduced nonradiative recombination loss by introducing PC$_{71}$BM, which can be widely observed in OSCs based on acceptors with the BTP central core[47]. The EQE spectra of the binary and ternary devices are shown in Fig. 4c. The two optimized devices both show efficient and broad photon-to-electron response range from 300 to 1050 nm, which is consistent with the absorption spectra of the blend films, and exhibited high EQE response in the NIR region. The integrated J$_{sc}$ from the EQE spectra of the binary and ternary devices are 27.83 and 28.31 mA cm$^{-2}$, respectively, which is in good agreement with the J$_{sc}$ values measured from the J–V curves within 2% mismatch.

**Table 1 Photovoltaic parameters of the optimal binary OSCs based on PTB7-Th:BTPV-4F (1:1.5, w/w) and ternary OSCs based on PTB7-Th:BTPV-4F (1:1.5:0.15, w/w/w) under the illumination of AM1.5 G, 100 mW cm$^{-2}$.**

| Active layer | $V_{oc}$ (V) | $J_{sc}$ (mA cm$^{-2}$) | FF (%) | PCE[a] (%) | Integrated $J_{sc}$ from EQE (mA cm$^{-2}$) |
|---|---|---|---|---|---|
| PTB7-Th:BTPV-4F | 0.65 (0.65 ± 0.01) | 28.3 (27.6 ± 0.6) | 65.9 (65.0 ± 1.1) | 12.1 (11.8 ± 0.3) | 27.83 |
| PTB7-Th:BTPV4F:PC$_{71}$BM | 0.67 (0.67 ± 0.01) | 28.9 (28.2 ± 0.5) | 69.3 (68.1 ± 0.9) | 13.4 (13.0 ± 0.2) | 28.31 |

[a]The average values were extracted from 30 devices.

To investigate the charge transfer dynamics of the active layer, femtosecond transient absorption spectroscopy (fs-TA) measurement was carried out on the pristine BTPV-4F film and the PTB7-Th:BTPV-4F blend film. Immediately following excitation of pristine BTPV-4F, we observed a broad ground state bleach (GSB) with a peak at 871 nm and two excited state absorption (ESA) signals at 610 nm and 990 nm (Supplementary Fig. 7). The decay of the signals showed a long excited state lifetime of 60 ps for exciton dissociation (The decay data of the excited state were best fit to an A → B → ground state (GS) kinetic model with time constants of $\tau_{A→B} = 3.3 ± 0.1$ ps and $\tau_{B→GS} = 58.4 ± 3.2$ ps (ca. 60 ps), so we think the lifetime is 60 ps). In the PTB7-Th:BTPV-4F blend film, after excitation at 850 nm, a GSB signal at 737 nm was apparent, which coincides with the absorption peak of PTB7-Th. This indicates the hole transfer from the acceptor BTPV-4F to the donor PTB7-Th. Fitting of the kinetic curves gave a fast charge transfer lifetime of 2 ps, indicating a high yield in charge generation and efficient photocurrent generation in the PTB7-Th:BTPV-4F systems.

Figure 4d compares the $J_{sc}$ values of the OSCs based on BTPV-4F and the devices based on other narrow bandgap acceptors reported in the literature (the corresponding photovoltaic parameters are listed in Supplementary Table 7). It can be seen that the BTPV-4F based device has the highest $J_{sc}$ (28.9 mA cm$^{-2}$) and BTPV-4F has the narrowest optical bandgap (1.21 eV) among all the reported works related to the narrow bandgap acceptors. Obviously, BTPV-4F exhibits great superiority in both optical bandgap and photovoltaic performance, which would be a promising acceptor for constructing the rear cell in high-performance tandem OSCs.

To further characterize the BTPV-4F based OSCs, charge carrier recombination behavior was studied by measuring the dependence of $J_{sc}$ vs. light intensity ($P_{light}$) (Supplementary Fig. 8). Generally, $J_{sc}$ and $P$ follow the relationship of $J_{sc} ∝ (P_{light})^α$. The α value should be 1 when there is no bimolecular recombination in the active layer[48]. For the BTPV-4F based OSCs, α values are calculated to be 0.975 and 0.998 for the binary and ternary devices respectively, indicating more efficient transportation of carriers and neglectable bimolecular recombination in the ternary device. The relationship between photocurrent density ($J_{ph}$) and effective voltage ($V_{eff}$) of the binary and ternary OSCs was further characterized to study the charge dissociation behavior of the devices[48]. As shown in Supplementary Fig. 9, under the short-circuit condition, the $J_{ph}/J_{sat}$ ratios are 0.982 for the binary device and 0.986 for the ternary device; under maximal power output conditions, the ratios are 0.839 and 0.858, respectively. The results indicate the ternary devices have the higher both exciton dissociation and charge collection efficiency.

The charge carrier mobilities of the active layers were estimated using the space charge limited current (SCLC) method (shown in Supplementary Fig. 10). The hole and electron mobilities ($\mu_h$ and $\mu_e$) of PTB7-Th:BTPV-4F binary systems are estimated to be ($9.4 ± 0.6) × 10^{-4}$ cm$^2$ V$^{-1}$ s$^{-1}$ and ($7.5 ± 0.7) × 10^{-4}$ cm$^2$ V$^{-1}$ s$^{-1}$, and the best values are $1.1 × 10^{-3}$ cm$^2$ V$^{-1}$ s$^{-1}$ and $8.3 × 10^{-4}$ cm$^2$ V$^{-1}$ s$^{-1}$, respectively. As for the PTB7-Th:BTPV-4F:PC$_{71}$BM ternary system, the corresponding hole and electron mobilities are ($9.2 ± 0.4) × 10^{-4}$ cm$^2$ V$^{-1}$ s$^{-1}$ and ($9.1 ± 0.6) × 10^{-4}$ cm$^2$ V$^{-1}$ s$^{-1}$, and the best values

are $1.0 × 10^{-3}$ cm$^2$ V$^{-1}$ s$^{-1}$ and $9.8 × 10^{-4}$ cm$^2$ V$^{-1}$ s$^{-1}$, respectively. The increased electron mobilities in ternary blends could be accountable for the higher FF in the ternary devices.

To study the morphology of the active layers, Grazing-incidence wide-angle X-ray scattering (GIWAXS) was conducted to determine the molecular packing of the BTPV-4F based films. Two-dimensional (2D) patterns and line-cut profiles for neat films are shown in Supplementary Fig. 11. For the neat BTPV-4F film, its π–π stacking (010) peak is at 1.82 Å$^{-1}$, corresponding to π–π stacking distance of 3.45 Å. Figure 5a, b shows the 2D GIWAXS patterns of the PTB7-Th:BTPV-4F blend film and PTB7-Th:BTPV-4F:PC$_{71}$BM blend film. (100) peaks are observed in the in-plane direction and (010) peaks are found in the out-of-plane (OP) direction (Supplementary Fig. 12), which suggests both films exhibit the more preferred face-on molecular packing orientation. With the addition of PC$_{71}$BM into the PTB7-Th:BTPV-4F blend, the π–π stacking distance is decreased, and both the integrated intensity and the coherence length of the OP π–π peak is increased (Supplementary Table 8). All these characteristics are associated with improved molecular ordering behavior of the ternary blend film, leading to higher charge carrier mobility thereby higher photovoltaic efficiencies.

The phase separation morphology of the blend films was also studied by atomic force microscopy (AFM). As shown in Fig. 5c, d, the AFM height images show a root-mean-square (RMS) roughness of 1.58 nm for the binary film and 1.31 nm for the ternary films. A smoother surface could favorably improve the contact between the interfacial layer and the active layer. We also measured the AFM phase images and TEM images. As shown in Supplementary Fig. 13, binary and ternary blend films showed a similar degree of phase segregation.

**Photovoltaic performance of tandem OSCs.** Finally, the optimized rear sub-cell was used in fabricating organic monolithic inverted-structured tandem solar cells. According to the analysis of the single-junction device result, PTB7-Th:BTPV-4F:PC$_{71}$BM system was used as active layer materials in rear cells. To ensure the ideal match of $J_{sc}$ with the rear cell, PM6:m-DTC-2F was used as a front cell active layer material. As shown in Supplementary Fig. 14, the absorption of PM6:m-DTC-2F films covers the short wavelength region from 300 nm to 770 nm, which is complementary to the absorption of the rear cell. Additionally, the PM6:m-DTC-2F based single-junction devices exhibit a high $V_{oc}$ of ~ 1 V and a high EQE response, giving a current of 17.1 mA cm$^{-2}$ as shown in Supplementary Fig. 15 and Supplementary Table 9–12. The structure of the tandem cell and corresponding energy levels diagram are shown in Fig. 6a, b. After the optimization of device performance, the best tandem OSC achieved a high PCE of 16.4% with a $V_{oc}$ of 1.65 V, a $J_{sc}$ of 14.5 mA cm$^{-2}$, and FF of 0.685, as shown in Fig. 6c and Table 2. The $V_{oc}$ of the tandem solar cell is 1.65 V, which is only 0.01 V less than the sum of the $V_{oc}$ values of the two sub-cells, indicating a well-formed effective tunneling junction in the interconnection layer. In this work, m-DTC-2F has a $V_{oc}$ loss of ca. 0.61 eV, which is one of the lowest values in the reported materials with a similar optical bandgap of around 1.6 eV. However, in the current stage of tandem OSCs, higher percentage of

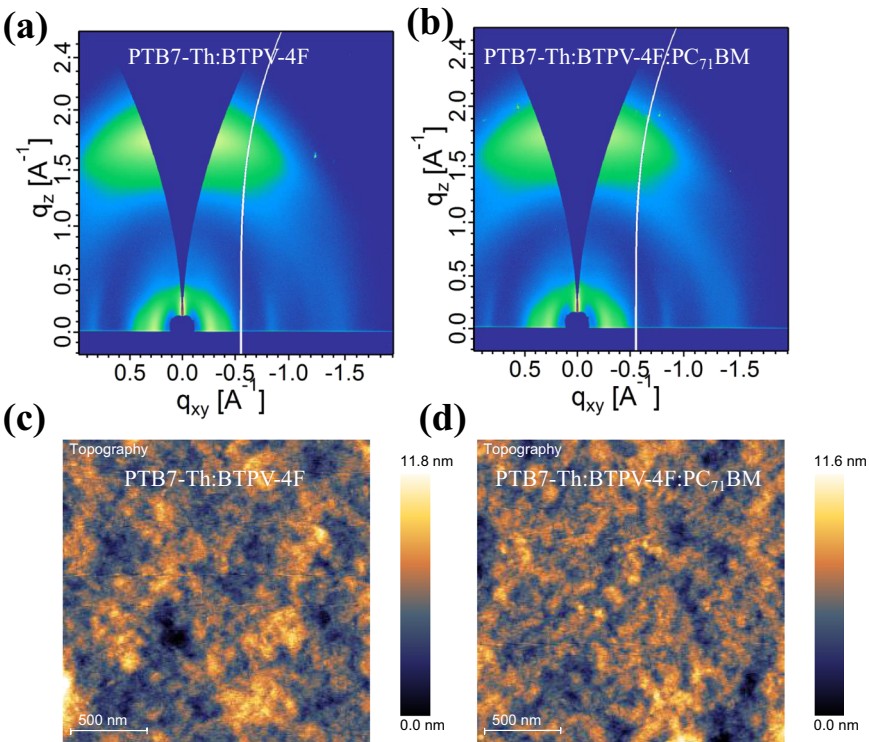

**Fig. 5 GIWAXS and AFM analysis of PTB7-Th:BTPV-4F and PTB7-Th:BTPV-4F:PC$_{71}$BM films.** 2D GIWAXS patterns of (**a**) binary blend film of PTB7-Th:BTPV-4F and (**b**) ternary blend film of PTB7-Th:BTPV-4F:PC$_{71}$BM. AFM images of (**c**) PTB7-Th:BTPV-4F and (**d**) PTB7-Th:BTPV-4F:PC$_{71}$BM films.

$V_{oc}$ loss still originates from the wide bandgap materials in front cells. Future works could focus more on the development of high-performance wide bandgap materials to further boost the efficiency of tandem OSCs.

To understand the current matching behavior between front and rear sub-cells in the tandem OSCs, EQE spectra of the sub-cells were measured, as shown in Fig. 6d. In monolithic tandem OSCs, the EQE spectra of one specific sub-cell are acquired by applying bias illumination to the other sub-cell to a level that the under-investigated sub-cell is current-limiting in its own absorption spectrum range. In this study, the light bias was obtained by a 550 nm short wave pass filter and a 850 nm long wave pass filter, for the measurements of the rear and front cell, respectively. The rear cell shows a broad EQE spectrum with a high response over 65% in the wavelength range of 760–970 nm. The integrated $J_{sc}$ of front and rear sub-cells are 14.27 and 14.32 mA cm$^{-2}$, respectively, indicating the highly balanced current generation in each sub-cell. The high and well-balanced $J_{sc}$ of the tandem OSCs is mainly attributed to the complementary absorption range of both sub-cells and the carefully tuned thickness of each active layer. The detailed thickness dependence of photovoltaic performance is listed in Supplementary Table 13. More importantly, the fabrication process of the rear cell was greatly facilitated by the benevolent morphology and the appropriate degree of phase segregation of BTPV-4F, and our devices showed excellent reproducibility as shown in the statistical diagram in Fig. 6e.

The optical light-field simulations were carried out to determine the optimal thicknesses of the sub-cells in experiments[49], as shown in Fig. 7a. The refractive index $n$ and extinction coefficient $k$ of all the layers used in the tandem OSCs were measured by a spectroscopic ellipsometer. Based on the optical simulation, an theoretically optimal value of over 14.8 mA cm$^{-2}$ was obtained when the optimal thicknesses of the active layers in the front and rear cells were 138 and 99 nm, respectively. The simulation result of the optimal thickness is consistent with our experimental results in the *J–V* measurements.

Long-term photostability is critical for the commercialization of this photovoltaic technology[50]. To date, very few works studied the photostability of tandem OSCs. Considering the commonly used light-emitting diode (LED) lamps fail to completely cover the infrared region of the absorption spectrum of the rear cell, a solar simulator with a metal halide lamp was used as the testing light source with illumination intensity equivalent to 1 sun. The illumination spectrum of the light source is included in Supplementary Fig. 16. Device photostability under continuous illumination for the tandem devices and single-junction devices is shown in Fig. 7b and the decay evolution of photovoltaics parameters of the three kinds of devices are shown in Supplementary Fig. 17. Encouragingly, the PTB7-Th:BTPV-4F:PC$_{71}$BM device shows excellent stability with 91% PCE remaining after 500 h aging. This result indicates that introducing double bond into non-fullerene acceptors is an advantageous way of constructing stable-structured low bandgap molecules with good photostability. However, the front cell based on PM6:*m*-DTC-2F shows obvious burn-in from the beginning and PCE just remains 40% after 500 h aging. Interestingly, thanks to the excellent stability of rear sub-cell, the stability of total tandem OSC is much improved even with the relatively unstable front sub-cell and the $T_{80}$ lifetime is over 200 h. The relatively weak light distribution would surely show lower decay rate of sub-cells, which causes the tandem devices to exhibit better overall light stability over single-junction sub-cells. It is reasonable to conclude that the tandem structure is superior in improving photostability of OSCs, and a more stable front cell materials in the future would lead to significantly more enhanced tandem OSCs stability.

**Discussion**

In summary, we introduced one more double bond π-bridge into the A-DA'D-A structured non-fullerene acceptor Y6 and

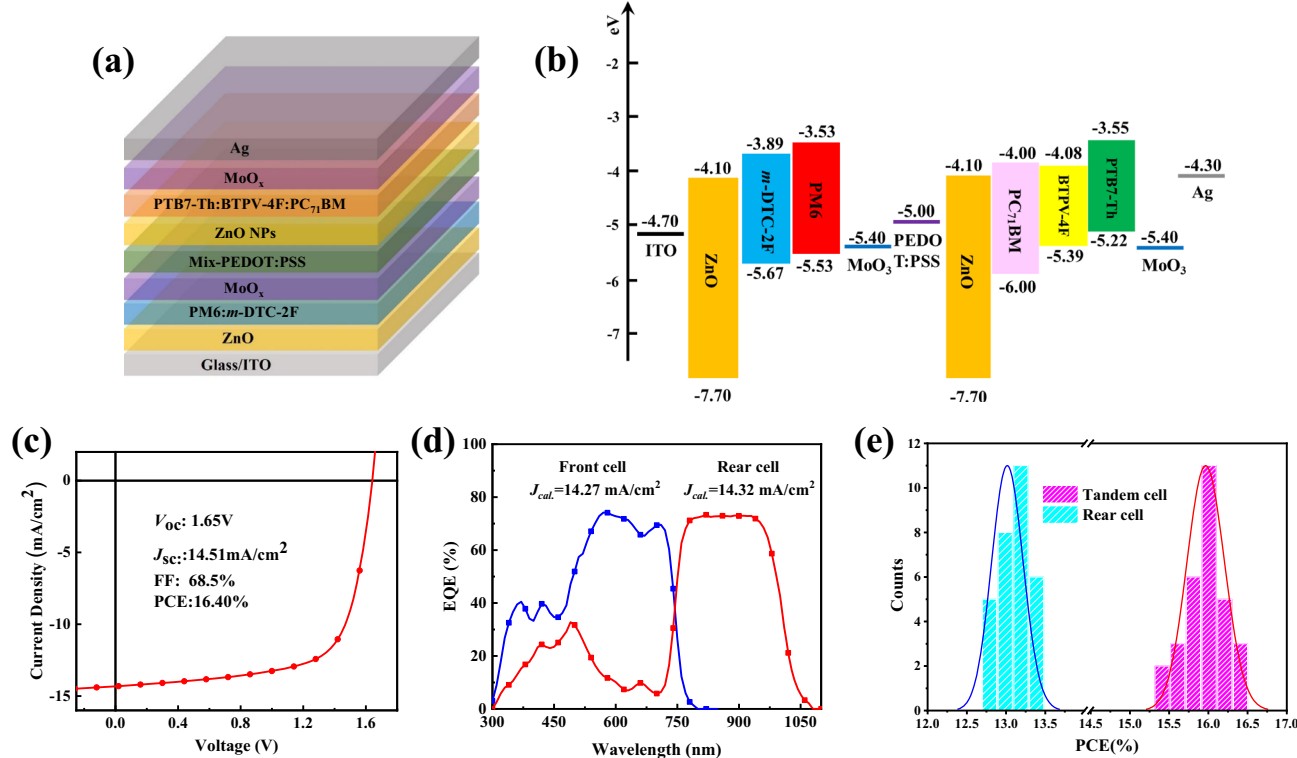

**Fig. 6 Device structure and photovoltaic performance of tandem solar cells. a** Device architecture of the tandem OSCs. **b** Energy level diagram of the related materials in the tandem OSCs. **c** The *J–V* curves of the optimal tandem OSC under the illumination of AM 1.5 G, 100 mW cm$^{-2}$. **d** EQE spectra of the front and rear cells of the tandem OSC. **e** PCE distribution of the BTPV-4F based single-junction OSCs and tandem OSCs based on 30 individual devices.

**Table 2 Photovoltaic parameters of the optimal single-junction OSCs and tandem OSC.**

| Device | $V_{oc}$ (V) | $J_{sc}$ (mA cm$^{-2}$) | FF (%) | PCE$_{max}$ (%) |
|---|---|---|---|---|
| Front cell[a] | 0.99 (0.99 ± 0.01) | 17.4 (17.0 ± 0.3) | 68.3 (67.3 ± 1.2) | 12.1 (11.8 ± 0.2) |
| Rear cell[b] | 0.67 (0.67 ± 0.01) | 28.9 (28.0 ± 0.4) | 69.3 (68.1 ± 0.9) | 13.4 (13.0 ± 0.2) |
| Tandem device | 1.65 (1.65 ± 0.01) | 14.5 (14.3 ± 0.3) | 68.5 (67.7 ± 1.1) | 16.4 (15.9 ± 0.3) |

[a]The thickness of the active layer is 140 nm.
[b]The thickness of the active layer is 100 nm.
[c]The average values were extracted from 30 devices.

synthesized a new ultra-narrow bandgap acceptor BTPV-4F. BTPV-4F shows significantly redshifted absorption with an $E_g^{opt}$ of 1.21 eV, because of its extended molecular conjugation length. The OSC based on PTB7-Th:BTPV-4F:PC$_{71}$BM (1:1.5:0.15, w/w/w) demonstrated a record $J_{sc}$ of 28.9 mA cm$^{-2}$ and overall PCE of 13.4%, with a broad EQE response wavelength range from 300 to 1050 nm. BTPV-4F shows a high potential for application in a rear sub-cell of tandem OSCs. Then we synthesized a medium bandgap acceptor *m*-DTC-2F with a bandgap of 1.61 eV, and select PM6: *m*-DTC-2F as the front sub-cell active layer for well-matching absorption of the front and rear sub-cells to fabricate the tandem OSCs. Due to the efficient utilization of solar spectrum in the range of 300−1050 nm and the reduced energy losses of two sub-cells, high PCE of 16.4% is achieved for the tandem OSC, which is one new case among all the best-performing tandem PSCs to date. we believe that the PCE of tandem OSCs can surpass 18% if FF can be promoted to over 75% by optimizing front cell material systems. So, in the next step we should design high-performance front cell material systems with high FF (ca. 75%) in thick film device and low energy loss (ca. 0.55 eV). Furthermore, the BTPV-4F based OSC shows excellent photostability with 91% PCE remaining after 500 h aging. All of the results indicate that inserting the double bonds π-bridge into the A-DA'D-A organic semiconductor acceptors is a simple and effective way to construct NIR organic acceptors with high performance and stability.

## Methods

**Materials**. The polymer donor PM6 and [6,6]-Phenyl-C71-butyric acid methyl ester (PC$_{71}$BM) were purchased from Solarmer Materials Inc. The polymer donor PTB7-Th was purchased from 1-Material Inc. BTP-CHO was purchased from Derthon Optoelectronic Materials Co., Ltd. Other chemicals and solvents were obtained from J&K, Energy Chemical, and Sigma Aldrich Chemical Co., respectively. All of the reagents and commercial compounds were used as received. The synthetic routes of acceptor BTPV-4F and *m*-DTC-2F are shown in Fig. 2 and Fig. 3, respectively, and the detailed synthesis processes are described in the following.

**Synthesis of BTPV-CHO**. To a solution of the mixture of BTP-CHO (205 mg, 0.2 mmol) and tributyl(1,3-dioxolan-2-ylmethyl)phosphonium bromide (94 mg, 0.22 mmol) in anhydrous tetrahydrofuran was added sodium hydride (60% dispersed in mineral oil, 0.6 mmol) under an argon gas atmosphere, and the resulting turbid solution was stirred at room temperature for 16 h. After completion of the reaction, the reaction was quenched using a 10% HCl solution under cooling and the

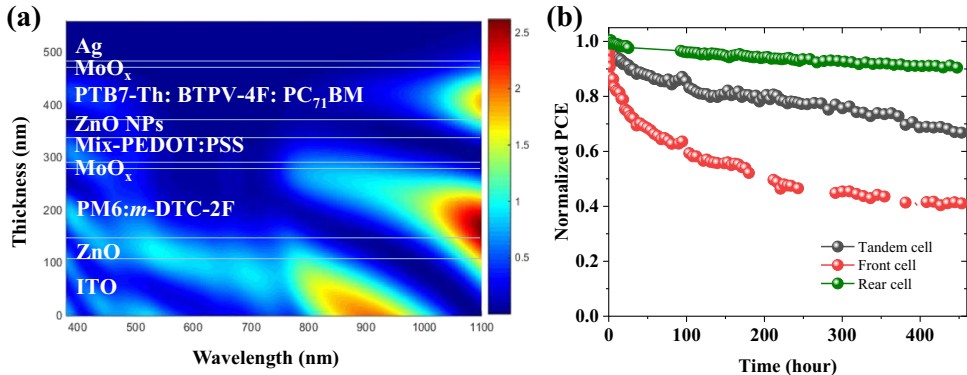

**Fig. 7 Light-field analysis and and long-term photostability test. a** The light-field distribution of the tandem OSC (140 nm for front sub-cell and 100 nm for rear sub-cell). **b** Long-term photostability test for the tandem solar cells and two sub-cells under continuous AM1.5 100 mW cm$^{-2}$ illumination.

reaction mixture was brought to acidic pH and stirred at room temperature for 4−5 h. The contents of the reaction flask were concentrated, and the organic contents were extracted into ethyl acetate. The organic layer was washed with water followed by brine, dried with anhydrous MgSO$_4$, filtered, and evaporated to dryness to afford the crude aldehyde that after purification by silica gel column chromatography using dichloromethane gave the product as a red solid (162 mg, 75% yield). $^1$H NMR (400 MHz, CDCl$_3$) δ (p.p.m.): 9.73 (d, $J = 7.8$ Hz, 2H), 7.78 (d, $J = 15.2$ Hz, 2H), 6.53 (dd, $J = 15.2$, 7.8 Hz, 2H), 4.61 (d, $J = 7.8$ Hz, 4H), 3.00 (t, $J = 7.6$ Hz, 4H), 2.12–1.96 (m, 2H), 1.86 (t, $J = 7.6$ Hz, 4H), 1.22–1.61 (m, 35H), 0.81–1.15 (m, 23H), 0.49–0.71 (m, 12H). $^{13}$C NMR (101 MHz, Chloroform-$d$) δ (p. p.m.) 192.48, 147.52, 143.72, 142.62, 142.06, 136.89, 133.30, 132.48, 125.79, 125.40, 125.14, 112.21, 55.38, 40.15, 31.91, 29.88, 29.70, 29.65, 29.61, 29.55, 29.43, 29.33, 28.38, 27.62, 27.60, 23.16, 22.68, 14.11, 13.68, 10.13, 10.11.

**Synthesis of BTPV-4F.** BTPV-CHO (162 mg, 0.15 mmol), 2-(5, 6-difluoro-3-oxo-2,3-dihydro-1H-inden-1- ylidene)malononitrile (140 mg, 0.60 mmol), pyridine (0.5 mL) and chloroform (20 mL) were dissolved in a round bottom flask under nitrogen. The mixture was stirred at room temperature overnight. Then, the mixture was purified with column chromatography using dichloromethane to give BTPV-4F (164 mg, 73% yield). $^1$H NMR (400 MHz, CDCl$_3$): δ (p.p.m.) 8.65–8.47 (m, 6H), 7.87–7.58 (m, 4H), 4.67 (d, $J = 8.0$ Hz, 4H), 3.02 (t, $J = 7.8$ Hz, 4H), 2.12-1.96 (m, 2H), 1.86 (t, $J = 7.6$ Hz, 4H), 0.52–1.51 (m, 66H) $^{13}$C NMR (101 MHz, Chloroform-$d$): δ (p.p.m.) 187.21, 157.40, 147.50, 146.85, 145.19, 145.03, 144.32, 137.69, 136.98, 136.62, 134.94, 132.80, 127.82, 123.42, 122.22, 115.06, 114.84, 114.36, 113.15, 112.51, 77.22, 69.46, 55.38, 40.17, 31.92, 30.18, 29.75, 29.69, 29.65, 29.62, 29.55, 29.47, 29.34, 28.67, 27.72, 27.70, 23.24, 22.76, 22.69, 14.11, 13.72, 10.28, 10.26. HRMS (TOF) $m/z$ calcd. for [M]$^+$ C$_{86}$H$_{90}$F$_4$N$_8$O$_2$S$_5$ 1502.5726, found 1503.5757.

**Synthesis of *m*-DTC-2F.** The intermediate compound 1 was prepared according to the literature[51]. Compound 1 (280 mg, 0.2 mmol) and 2-(5 or 6-difluoro-3-oxo-2,3-dihydro-1H-inden-1-ylidene) malononitrile (150 mg, 0.7 mmol) were dissolved in CHCl$_2$ (25 mL) under a nitrogen atmosphere. 0.6 mL pyridine was added and refluxed for 12 h. Then the mixture was purified using column chromatography on silica gel employing petroleum ether/CHCl$_2$ (1:1) as an eluent, yielding a dark blue solid *m*-DTC-2F (245 mg, 69% yield). $^1$H NMR (400 MHz, Chloroform-$d$) δ 8.91 (d, $J = 3.4$ Hz, 2H), 8.71 (dd, $J = 8.8$, 4.2 Hz, 1H), 8.38 (dd, $J = 9.1$, 2.0 Hz, 1H), 8.04–7.80 (m, 4H), 7.71 (d, $J = 5.7$ Hz, 3H), 7.57 (dd, $J = 6.7$, 2.5 Hz, 1H), 7.42 (td, $J = 8.3$, 2.2 Hz, 2H), 7.21–6.90 (m, 16H), 4.62 (s, 1H), 2.52 (t, $J = 7.7$ Hz, 8H), 2.32 (dd, $J = 9.6$, 4.6 Hz, 2H), 2.12 (d, $J = 12.0$ Hz, 2H), 1.55-1.48 (m, 8H), 1.34–1.18 (m, 48H), 0.83-0.77 (m, 18H). $^{13}$C NMR (101 MHz, Chloroform-$d$) δ 187.18, 187.00, 168.00, 165.45, 161.60, 161.51, 159.08, 158.12, 147.22, 144.12, 144.09, 143.37, 142.35, 142.25, 140.61, 140.13, 140.05, 139.96, 138.74, 134.61, 134.15, 133.14, 128.39, 128.12, 127.30, 125.82, 125.73, 124.99, 124.42, 121.80, 121.56, 121.46, 121.39, 118.75, 118.43, 114.51, 114.36, 114.32, 112.95, 112.69, 104.91, 102.34, 77.33, 77.22, 77.01, 76.70, 69.61, 62.99, 57.58, 35.99, 33.67, 31.75, 31.62, 31.44, 29.40, 29.34, 29.15, 28.93, 27.17, 22.59, 14.06, 14.04. HRMS (TOF) $m/z$ calcd. for [M]$^+$ C$_{113}$H$_{117}$F$_2$N$_5$O$_2$S$_2$ 1677.8460.5726, found 1678.8622.

**Material characterization.** The NMR spectra were measured using Bruker AVANCE400 MHz spectrometer. Electrochemical measurements were carried out under nitrogen in a solution of tetra-$n$-butylammonium hexafluorophosphate ([$^n$Bu$_4$N]$^+$[PF$_6$]$^-$) (0.1 M) in CH$_3$CN employing a computer-controlled CHI660C electrochemical workstation, glassy carbon working electrode coated with films, an Ag/AgCl reference electrode, and a platinum-wire auxiliary electrode. The potentials were referenced to a ferrocenium/ferrocene (FeCp$_2$$^{+/0}$) couple using ferrocene as an internal standard.

**Device fabrication of single-junction OSCs.** The inverted structure device of PM6:*m*-DTC-2F was fabricated with the device architecture of ITO/ZnO/PM6:*m*-DTC-2F/MoO$_3$/Ag. The ZnO precursor solution is prepared by dissolving 0.14 g of zinc acetate dehydrate (99.9%, Aldrich) and 0.5 g of ethanolamine (NH$_2$CH$_2$CH$_2$OH, 99.5%, Aldrich) in 5 ml of 2-methoxyethanol (CH$_3$OCH$_2$-CH$_2$OH, 99.8%, J&K). A thin layer of ZnO is deposited through spin-coating the ZnO precursor solution on precleaned ITO glass at 5000 rpm and baked subsequently at 200 °C for 2 h. Then the device is transferred into a glove box filled with nitrogen, in which the active layer of PM6:*m*-DTC-2F (1:1.5 w/w) (22 mg/mL in total, from CB with 0.3% DIO) is spin-coated onto the ZnO layer at 1500 rpm. After that, the active layer is annealed at 110 °C for 10 min for the devices with thermal annealing treatment. The thickness of the active layer is ca. 100 nm. Finally, a layer of ca. 10 nm MoO$_3$ and then a Ag layer of ca. 100 nm is evaporated subsequently under high vacuum.

The inverted structure device of PTB7-Th:BTPV-4F or PTB7-Th:BTPV-4F: PC$_{71}$BM was fabricated with device architecture of ITO/ZnO/active layer/MoO$_3$/ Ag. The ZnO precursor solution is prepared by dissolving 0.14 g of zinc acetate dehydrate (99.9%, Aldrich) and 0.5 g of ethanolamine (NH$_2$CH$_2$CH$_2$OH, 99.5%, Aldrich) in 5 ml of 2-methoxyethanol (CH$_3$OCH$_2$CH$_2$OH, 99.8%, J&K). A thin layer of ZnO is deposited through spin-coating the ZnO precursor solution on precleaned ITO glass at 5000 rpm and baked subsequently at 200 °C for 2 h. Then the device is transferred into a glove box filled with nitrogen, in which the active layer of PTB7-Th:BTPV-4F:PC$_{71}$BM (1:1.5:0.15 w/w/w) (16 mg/mL in total, from CF with 0.5% CN) is spin-coated onto the ZnO layer at 3000 rpm. After that, the active layer is annealed at 110 °C for 5 min for the devices with thermal annealing treatment. The thickness of the active layer is ca. 100 nm. Finally, a layer of ca. 10 nm MoO$_3$ and then a Ag layer of ca. 100 nm is evaporated subsequently under high vacuum.

**Device fabrication of tandem devices.** The tandem OSCs were fabricated with an architecture of ITO/ZnO/ PM6:*m*-DTC-2F /MoO$_3$/M-PEDOT/ZnO NPs/PTB7-Th:BTPV-4F:PC$_{71}$BM /MoO$_3$/Ag. The PM6:*m*-DTC-2F active layers were fabricated via the same process as the single cells with different thicknesses. Subsequently, 10 nm MoO$_3$ was evaporated under high vacuum. M-PEDOT:PSS solution was prepared by mixing PEDOT:PSS with 1 vol% nonionic ethoxylated fluorosurfactant (Zonyl FSO). The M-PEDOT:PSS solution was then spin-casted (5,000 rpm) and annealed at 130 °C for 2 min. Then, ZnO nanoparticles in IPA (20 nm) was spin-coated and annealed at 120 °C for 5 min in the glove box. Then, the PTB7-Th:BTPV-4F:PC$_{71}$BM active layers were fabricated via the same process as the single-junction devices with different thicknesses. Finally, a layer of 10 nm MoO$_3$ and then a Ag layer of 100 nm is evaporated subsequently under high vacuum. The device area of the OSCs was 5.0 mm$^2$, which was defined by optical microscope (Olympus BX51). In order to accurately measure the photocurrent, mask with an area of 4.80 mm$^2$ was used to define the effective area of the OSCs. The devices with or without mask showed consistent photovoltaic performance values with relative errors within 0.3%.

The current density–voltage ($J$–$V$) characteristics of the OSCs were measured in a nitrogen glove box with a Keithley 2450 Source Measure unit. Class AAA Solar Simulator (SS-X100R, Enlitech) with a 450 W xenon lamp and an air mass (AM) 1.5 filter was used as the light source. Output spectrum of the solar simulator is provided in Supplementary Fig. 24. The light intensity was calibrated to 100 mW cm$^{-2}$ by a SRC-2020 reference cell with a KG2 filter window. The mismatch factor is calculated as 1.0048. The input photon to converted current efficiency (IPCE) was measured by Solar Cell Spectral Response Measurement System QE-R3-011 (Enli Technology Co., Ltd., Taiwan). The light intensity at each wavelength was calibrated with a standard single-crystal Si photovoltaic cell. To measure the rear and front cell, light bias

obtained by 550 nm short wave pass filters and 850 nm long wave pass filters were selected to excite (saturate) the front and rear cells, respectively.

**UV–visible absorption**. The UV–vis absorption spectra were measured on a Hitachi U-3010 UV–vis spectrophotometer. For the measurement of films, the acceptors and blend films were prepared by spin-coating the polymer chloroform solutions on quartz plates.

**Atomic force microscope (AFM)**. The atomic force microscope used is a Vista-Scope from Molecular Vista, Inc., operated in dynamic mode using commercial gold-coated silicon cantilevers (NCHAu) from Nanosensors using tapping mode.

**Grazing incident wide-angle X-ray scattering (GIWAXS)**. Grazing-incidence wide-angle X-ray scattering (GIWAXS) measurements were conducted at Advanced Light Source (ALS), Lawrence Berkeley National Laboratory, Berkeley, CA at the beamline 7.3.3. Data were acquired at the critical angle ($0.13^0$) of the film with a hard X-ray energy of 10 keV. X-ray irradiation time was 30–60 s, dependent on the saturation level of the detector. Beam center was calibrated using AgB powder and the sample-to-detector distance was about 280 mm. The π-π coherence lengths (L) are estimated based on the Scherrer Equation (L = 2πK/FWHM), where K is the shape factor (here we use 0.9), and FWHM is the full width at half maximum of the (010) diffraction peaks.

**Transient absorption spectroscopy**. A Yb: KGW amplifier (PHAROS,Light Conversion,) supplied laser beams centered at 1030 nm with pulse duration of ~180 fs, pulse repetition rate of 33 kHz, and a maximum pulse energy of 0.3 mJ. The output of the amplifier was split into two streams of pulses. One was used to drive an optical parametric amplifier (ORPHEUS-HP, Light Conversion) to obtain the pump beam. The residual stream was directed into an ultrafast spectroscopic system (HARPIA-TA, Light Conversion) to generate the white light continuum probe beam. In the spectrometer, the pump chopped at 150 Hz frequency was spatially and temporally overlapped with the probe beam on the sample. Pump wavelength was set to 850 nm to selectively excite the BTPV-4F acceptor. Excitation energy of the pump pulse was set to 2 μJ/cm$^2$ to avoid singlet-singlet annihilation. The film samples for TA measurements were prepared by spin coating the corresponding materials on quartz plates of 1 mm thick. The TA samples were annealed in nitrogen atmosphere at 100 °C for 3 min prior to measurement.

**Measurement of charge carrier mobilities**. The charge carrier mobilities were measured with the device structure of ITO/PEDOT:PSS/active layer/MoO$_3$/Ag for hole-mobility and ITO/ZnO/active layer/PDINO/Al for electron mobility. The hole and electron mobilities are calculated according to the SCLC method equation: $J = 9\mu\varepsilon_r\varepsilon_0 V^2/8d^3$, where $J$ is the current density, $\mu$ is the hole or electron mobility, $V$ is the internal voltage in the device, $\varepsilon_r$ is the relative dielectric constant of active layer material, $\varepsilon_0$ is the permittivity of empty space, and $d$ is the thickness of the active layer.

**Stability test**. Devices were sealed in glass fronted chambers under N$_2$ without moisture (water and oxygen < 0.5 ppm). Solar Simulator with a metal halide lamp was used as a light source with intensity equivalent to 1 sun, which was calibrated by matching the device performance to those measured under AM1.5 G with the ultraviolet portion removed by filtering and a Keithley 2400 source meter. The initial exposure time is defined as time 0 h. The temperature of the cells was kept under 40 °C by a fan during measurements.

**Reporting summary**. Further information on research design is available in the Nature Research Reporting Summary linked to this article.

## Data availability
The data that support the findings of this study are available from the corresponding author on request.

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

## Acknowledgements

This work was supported by the National Key Research and Development Program of China (No. 2019YFA0705900) funded by MOST, NSFC (Nos. 51820105003, 21734008, and 61904181) and the Guangdong Major Project of Basic and Applied Basic Research (No. 2019B030302007). X-ray data were acquired at beamline 7.3.3 at the Advanced Light Source, which is supported by the Director, Office of Science, Office of Basic Energy Sciences, of the U.S. Department of Energy under Contract No. DE-AC02-05CH11231.

## Author contributions

Z.J. synthesized and characterized the acceptors of BTPV-4F and *m*-DTC-2F, S.Q. and L.M. performed the device fabrication and characterization of the single-junction and tandem OSCs, X.L. participated in the molecular design and synthesis of the acceptors, Q.M. and W.L. participated in the characterizations of materials and devices, J.Z. measured and analyzed transient absorption spectroscopy, I.A. and H.A. measured and analyzed the GIWAXS results. Y.H., N.L., and C.J.B. measured and analyzed the device photostability. Y.L. and L.M. supervised the project, and Z.J., S.Q., L.M., and Y.L. wrote the paper.

## Competing interests

The authors declare no competing interests.
