## [Peer Review File · Nature Communications]

REVIEWER COMMENTS

Reviewer #1 (Remarks to the Author):

In this work, authors have developed narrow band gap acceptor BTPV-4F with an optical bandgap of 1.21 eV, through the extension of double bond bridge of Y6 acceptor with one more vinyl group. Interestingly, the single-junction OSCs based on BTPV-4F as acceptor have achieved power conversion efficiency (PCE) of over 13.4%, with a high short-circuit current density (J_{sc}) of 28.87 mA cm⁻². Further, tandem devices were fabricated based on BTPV-4F:PC71BM: PTB7-Th rear cell, and PM6:carbazole acceptor m-DTC-2F as front cell, which reached a fairly high PCE of 16.4% with good photostability. Overall, this work contains a good set data covering both material chemistry and device engineering. It would be very suitable for attracting the readership of journals. Therefore, reviewer would like to recommend this work can be accepted for the publication in Nature communications, after some technique concerns being addressed.

(1) The inserting of another ethylene double bond indeed broadens the absorption of NFAs, however, also introducing the high possibility for structural isomerism of molecules. Vinyl group is also liable to photoisomerization. Please verify the molecular structure of BTPV-4F in manuscript (scheme 1 and figure 1), also put NMR spectra into SI. it is important information.

(2) Dose the absorption ability of BTPV-4F become stronger over that of Y6? Please provide the extinction coefficient comparison between BTPV-4F and Y6 in solution and film.

(3) Not sure whether the devices are measured under aperture. It should be done so. Please provide the related information in experimental section.

(4) To accurate determine the PCE of tandem cells, mismatch factor is usually needed with information including the output spectrum of the solar simulator and the spectral response of the reference cell.

(5) Interestingly to note that the stability curve of the tandem device is ranging between of two single cells. Why the tandem device has better stability over that of single ones? The decay of any sub-cells would lead to decay of tandem cells.

(6) Please check the spelling carefully, especially the subscript and italic characters in manuscript and figures.

Reviewer #2 (Remarks to the Author):

The authors describe the synthesis of new narrow and wide band gap acceptor molecules for use in tandem solar cells. The paper is interesting. The authors show that by inserting additional double bonds it is possible to shift the optical band gap to lower energies and obtain an electron acceptor that can generate the largest photocurrent in a single-junction organic solar cell. They then combine it with a wide band gap cell to reach a highly efficient and fairly stable organic tandem solar cell. Although one may argue that inserting just a double bonds is not a big step, it shows that atom-by-atom optimization of known acceptor molecules can lead to materials with new and enhanced properties. Having said all this is, there are quite a number of smaller and bigger issues to be solved before publication can be considered.

1. The English needs improvement. The first sentence of the abstract two errors. Someone proficient in English should correct the entire text.

2. The number of significant digits given for several parameters such as currents, efficiencies and mobilities really exceeds the error margins. This should be looked at critically. As an example if one uses a 2 mm x 2 mm solar cell area and the J_{sc} is quoted as 28.00 mA/cm² i.e. with 4 digits as is done in the paper, the accuracy in the dimensions should be 2.0000 ± 0.0004 mm. This is unrealistic. That would also hold for the light intensity which would need also an accuracy of 0.04%.

3. Page 2, the sentence on lines 52-56 needs considerable improvement, both in its grammar and its meaning.

4. The authors frequently use infrared to refer to their spectral range. It is better to name this near infrared (NIR), which is the region up to 1400 nm.
5. On page 4 the authors argue that narrow bandgap materials are most needed for tandem OSCs. I do not necessarily agree with that. As most novel non-fullerene acceptors (NFAs) have quite narrow band gaps, there is much more need for wide band gap material. As I will discuss below, the performance of the tandem cell is limited by the wide band gap cell. This should be discussed in more detail and in fact, what I take a lot about this manuscript is the development of a wide bandgap NFA. In my view the authors can balance this discussion much better, not only focusing on the fact that both NFAs presented in this paper are important.
6. The DFT calculation give HOMO and LUMO levels for the new BTPV-4F acceptor and Y6, but the values are never compared to the experimental data. That should have been done.
7. The authors should more clearly define the method on how they determine the optical band gap. Looking at the optical spectrum of the BTPV-4F film I would judge its onset at a higher photon energy than 1050 nm. This is also because some of the highlight of the paper is derived from Figure 2d, and a shift of the band gap to higher energy would result in a less remarkable result.
8. Please also include in the text or Supp. Info the HOMO and LUMO levels of Y6, PM6, PTB7-Th, and PC71BM determined using the same methods, to give insight in the relative energy levels.
9. Why is the Voc enhanced using PC71BM?
10. Regarding the transient absorption I do not understand the lifetime of 60 ps. I judge it at about 20 ps. At the lifetime the intensity has decayed to $1/e$ ($e = \exp(1)$) and this happens at about 20 ps. How did the authors come to 60 ps? In addition, I would not consider 60 ps (and certainly not 20 ps) as a long lifetime. It is pretty short and will certainly the limit exciton diffusion length. Some more discussion here would be useful.
11. I do not see how a lower mobility can result in an increased fill factor (FF). The mobilities are quoted with three significant digits. Is that realistic? What are the error margins from the fits? Because the layer thickness goes with the power 3 in the SCLC equation an error (or variation) of 10 nm (see AFM data) (at 200 nm nominal thickness) already gives a 15% difference in the mobility. Also looking at the data in Supp. Fig. 10b the lower electron mobility of the blend with PC71BM seems to be due to two or three data points at lower voltages. Furthermore I would like to see more evidence that a change in the mobility ratio of 1.48 to 1.06 has an effect on the FF. From the paper of Koster et al, in DOI: 10.1038/ncomms8083 it is not obvious that better balance of mobilities increase the FF. It could also be just a geometrical effect due to a higher Voc. When a JV curve is shifted over the V-axis to higher Voc, the FF will be higher.
12. In the in-plane and out-of-plane GIWAXS data one sees a clear (100) peak of PTB7-Th in the neat film at 0.6 \AA^{-1} , why is that absent in the blends? Likewise the blend has a signal at 0.8 \AA^{-1} that is not in the pristine material.
13. Suppl. Table 8 does not provide the integrated intensities as suggested in the main text. Looking at Supp Fig. 12 I have hard time in seeing that the crystallinity increases. The text is also ambiguous there. It is stated that the higher crystallinity increases the mobility, but the authors actually showed earlier that the hole mobility decreases when P71BM is added. So I do not understand. The authors should also more clearly indicate which component we are main looking at in the GIWAXS. Is it PTB7-Th or BTPV-4F, or is it impossible to tell? This can be explained more clearly.
14. In the discussion of the AFM the lower surface roughness with PC71BM is used as a reason for higher charge carrier generation due to better mixing. But there is no evidence for that. The current increase when PC71BM is due to absorption by PC71BM as can be seen in the EQE data of Figure 2c. In the other spectra range there is no increase in EQE and hence there is not better charge generation.
15. Also the argument of an increase CCL being beneficial and better mixing being also beneficial seem at variance with each other. I know that all these arguments are frequently used in the literature, and mostly to show that they can be favorable. One could also state that if the lower surface roughness indicates better mixing the mobility would go down and that an increased CCL would limit charge generation. I think the authors should make a more balanced discussion.
16. What the absolute energy given the Fc/Fc+ couple to determine the HOMO and LUMO

energies?

17. Why is surfactant added to PEDOT:PSS for spin coating on a MoO₃ layer? Which is that surfactant?

18. The device area is missing, so is the use of any masks. Please state if they are used or not and what is their relative to the nominal device area. How was the device area determined and how accurate?

19. In the end I would like to come back to the discussion on which cell is now limiting more. The wide band gap cell has a voltage loss of $1.6 - 1.0 = 0.60$ eV, the narrow band gap cell has a voltage loss of $1.20 - 0.67 = 0.53$ eV. I think that at present the best OSCs have larger Voc losses for higher band gaps. This would actually argue that it is more important to develop wide bandgap sub cells than narrow band gap sub cells. A more detailed discussion on that could make this paper much more solid and important.

Reviewer #3 (Remarks to the Author):

This paper developed a new structured non-fullerene acceptor material that can absorb the IR region and applied it to organic solar cells. The development of high-performance photoactive materials capable of absorbing light in various wavelength regions is an important means of ultimately increasing the usefulness of organic solar cells, colorfully and transparently, so it can be said to be a meaningful study. Therefore, although the logic of introducing new materials used the existing method and the tandem device did not show the best performance at the present time, it is judged that there is enough novelty to be accepted on Nature Commun. However, the followings should be corrected and supplemented.

On page 11, the bleach peak at 713 nm in Sup Fig. 7(c) is claimed to be matched with the absorption peak of PTB7-Th. In order to explain this logic, in addition to the movement of holes from excited BTPV-4F to PTB7-Th's HOMO, explanation on how electrons can be introduced into PTB7-Th's LUMO should be added.

On page 14, the phase segregation of donor and acceptor was inferred from the AFM's Topo image, but it was insufficient to confirm the phase segregation from the Topo image. To be more accurate, additional data of AFM phase image and TEM image are required.

On page 13, it was explained that J_{ph}/J_{sat} indicates charge generation and exciton dissociation efficiency in Sup. Fig. 9, but it seems reasonable that the expression of charge collection is also included.

On page 10, the number of significant digits in Voc varies from data to data (0.65 V and 0.671 V), please use the same significant digit.

Sept. 24, 2020

Response to reviewers:

Response to Reviewer #1:

In this work, authors have developed narrow band gap acceptor BTPV-4F with an optical bandgap of 1.21 eV, through the extension of double bond bridge of Y6 acceptor with one more vinyl group. Interestingly, the single-junction OSCs based on BTPV-4F as acceptor have achieved power conversion efficiency (PCE) of over 13.4%, with a high short-circuit current density (J_{sc}) of 28.87 mA cm⁻². Further, tandem devices were fabricated based on BTPV-4F:PC71BM: PTB7-Th rear cell, and PM6:carbazole acceptor m-DTC-2F as front cell, which reached a fairly high PCE of 16.4% with good photostability. Overall, this work contains a good set data covering both material chemistry and device engineering. It would be very suitable for attracting the readership of journals. Therefore, reviewer would like to recommend this work can be accepted for the publication in Nature communications, after some technique concerns being addressed.

(1) The inserting of another ethylene double bond indeed broadens the absorption of NFAs, however, also introducing the high possibility for structural isomerism of molecules. Vinyl group is also liable to photoisomerzation. Please verify the molecular structure of BTPV-4F in manuscript (scheme 1 and figure 1), also put NMR spectra into SI. it is important information.

Response: We appreciate the reviewer's comment, and we have added NMR spectra of the intermediate and the acceptors in Supplementary information (Supplementary Fig. 18-23) of the revised manuscript.

As it is well known, the *cis-trans* isomerism of ethylene double bond can be distinguished by the coupling constant of vinyl hydrogens. Vicinal coupling constants are larger for *trans* (range: 12–18 Hz; typical: 15 Hz) than *cis* (range: 0–12 Hz; typical: 8 Hz) isomers. From the ¹H NMR

spectra, BTPV-CHO shows the signals at 7.78 ppm (d, J= 15.2Hz, 2H) and 6.53 ppm (dd, J=15.2, 7.8 Hz, 2H), which belongs to the vinyl hydrogens on the inserted double bond. Due to the higher vicinal coupling constant, we consider that the introduced vinyl group in the acceptor should be determined as *trans* form, as shown in Scheme 1 and Figure 1.

In addition, our group had reported an acceptor called SJ-IC synthesized by inserting double bond to the IDT-IC (*J. Mater. Chem. A*, 2017, 5, 22588-22597) and we adopted the same synthetic route in this work. As shown in the following Figure R1, the single crystal structure of SJ-IC can also prove the *trans* form of introduced vinyl groups in the acceptor.

Figure R1. Crystal structure of the SJ-IC.

(2) Dose the absorption ability of BTPV-4F become stronger over that of Y6? Please provide the extinction coefficient comparison between BTPV-4F and Y6 in solution and film.

Response: The extinction coefficient of BTPV-4F is higher than Y6. We added the result and related discussion in p. 7: “The extinction coefficients of BTPV-4F and Y6 in chloroform solution are $1.52 \times 10^5 \text{ L mol}^{-1} \text{ cm}^{-1}$ and $1.31 \times 10^5 \text{ L mol}^{-1} \text{ cm}^{-1}$, respectively. The extinction coefficients of BTPV-4F and Y6 films are $1.21 \times 10^5 \text{ cm}^{-1}$ and $1.09 \times 10^5 \text{ cm}^{-1}$, respectively. BTPV-4F shows higher extinction coefficients in both solution and film than Y6 (Supplementary Fig. 4).” The absorption spectra of BTPV-4F and Y6 in chloroform solution and in films were compared and added in Supplementary Fig. 4.

(3) *Not sure whether the devices are measured under aperture. It should be done so. Please provide the related information in experimental section.*

Response: The related information of device area is added in method section in p. 26: “The device area of the OSCs was 5.0 mm², which was defined by optical microscope (Olympus BX51). In order to accurately measure the photocurrent, mask with an area of 4.80 mm² was used to define the effective area of the OSCs. The devices with or without mask showed consistent photovoltaic performance values with relative errors within 0.3%.”

(4) *To accurate determine the PCE of tandem cells, mismatch factor is usually needed with information including the output spectrum of the solar simulator and the spectral response of the reference cell.*

Response: Output spectrum of the solar simulator is provided in **Supplementary Fig 24**. To guarantee a full coverage of the response region of tandem solar cells, a reference silicon cell with a KG-2 filter window was selected. The light intensity was calibrated to 100 mW cm⁻² by the reference cell. The mismatch factor is calculated to be 1.0048.

(5) *Interestingly to note that the stability curve of the tandem device is ranging between of two single cells. Why the tandem device has better stability over that of single ones? The decay of any sub-cells would lead to decay of tandem cells.*

Response: The decay of any sub-cell would indeed lead to the decay of the tandem cell. In tandem devices, under the same 1 sun irradiance, the light intensity in each sub-cell is lower than that in single cells, due to the re-distributed light field. It is the same reason that the photo current of the sub-cell is much lower than that in single junction cells. The relatively weak light

distribution would surely show lower decay rate of sub-cells, which causes the tandem devices to exhibit better overall light stability over single junction sub-cells. The same discussion was added into p. 21: “The relatively weak light distribution would surely show lower decay rate of sub-cells, which causes the tandem devices to exhibit better overall light stability over single junction sub-cells”.

(6) Please check the spelling carefully, especially the subscript and italic characters in manuscript and figures.

Response: We appreciate the reviewer’s comment. We carefully checked the spelling issue.

Response to Reviewer #2

The authors describe the synthesis of new narrow and wide band gap acceptor molecules for use in tandem solar cells. The paper is interesting. The authors show that by inserting additional double bonds it is possible to shift the optical band gap to lower energies and obtain an electron acceptor that can generate the largest photocurrent in a single-junction organic solar cell. They then combine it with a wide band gap cell to reach a highly efficient and fairly stable organic tandem solar cell. Although one may argue that inserting just a double bonds is not a big step, it shows that atom-by-atom optimization of known acceptor molecules can lead to materials with new and enhanced properties. Having said all this is, there are quite a number of smaller and bigger issues to be solved before publication can be considered.

1. *The English needs improvement. The first sentence of the abstract two errors. Someone proficient in English should correct the entire text.*

Response: Thanks for the reviewer’s comments. We have checked the English of the manuscript

carefully and corrected the errors.

- 2. The number of significant digits given for several parameters such as currents, efficiencies and mobilities really exceeds the error margins. This should be looked at critically. As an example if one uses a 2 mm x 2 mm solar cell area and the J_{sc} is quoted as 28.00 mA/cm² i.e. with 4 digits as is done in the paper, the accuracy in the dimensions should be 2.0000 ± 0.0004 mm. This is unrealistic. That would also hold for the light intensity which would need also an accuracy of 0.04%.*

Response: We thank the reviewer for pointing out the accuracy issue of the data. The device area was determined by an aperture with area of 4.80 mm² (2.00 mm x 2.40 mm). The area of mask was measured under a camera integrated on the optical microscope (Olympus BX51) with error margin of 1.8 μm. Considering that the amount of significant digits in the final product can only have as many significant digits as the multiplicand with the least amount of significant digits, we change the amount of significant digits to three. In this case, for example, the J_{sc} should be 28.0 mA/cm². On the other hand, the NEWPORT solar simulator has a temporal instability less than 0.5%. Accordingly, all the other parameters were modified throughout the entire manuscript with three significant digits.

- 3. Page 2, the sentence on lines 52-56 needs considerable improvement, both in its grammar and its meaning.*

Response: For more clearly expressing the meaning, we revised the sentence into: “**While in the series-connected tandem OSCs, the absorption spectrum wavelength region can be effectively extended by employing a wide bandgap sub-cell to harvest high energy photons and another narrow bandgap sub-cell for utilizing low energy photons. At the same time, the open-circuit voltage (V_{oc}) of the tandem OSCs is the summation of those of the two sub-cells**”. (see p. 3 in the

revised manuscript.)

4. *The authors frequently use infrared to refer to their spectral range. It is better to name this near infrared (NIR), which is the region up to 1400 nm.*

Response: We have revised the words of “infrared” to “**near-infrared**” (NIR) in the revised manuscript.

5. *On page 4 the authors argue that narrow bandgap materials are most needed for tandem OSCs. I do not necessarily agree with that. As most novel non-fullerene acceptors (NFAs) have quite narrow band gaps, there is much more need for wide band gap material. As I will discuss below, the performance of the tandem cell is limited by the wide band gap cell. This should be discussed in more detail and in fact, what I like a lot about this manuscript is the development of a wide bandgap NFA. In my view the authors can balance this discussion much better, not only focusing on the fact that both NFAs presented in this paper are important.*

Response: We appreciate the reviewer’s suggestion. In recent years, many important narrow bandgap NFAs were acquired in this field and offered much higher single-junction efficiency than before. However, highly efficient narrow bandgap NFAs with absorption wavelength range extending beyond 1050 nm are extremely rare, which makes the efficiency of current-stage tandem OSCs less competitive. The high performance of the rear cell is prerequisite for the tandem OSCs, since it determines the eventual achievable J_{sc} of a tandem OSC. BTPV-4F reported in this work is by far the most applicable narrow bandgap acceptor with well-balanced absorption and efficiency. And we put our main efforts on the development strategy and characterization of narrow bandgap NFA BTPV-4F.

On the other hand, we also totally agree with the reviewer's opinion on wide bandgap materials. Indeed, for the improvement of tandem OSCs, the wide bandgap materials are as important as the NIR materials, because they determine the major part of the overall V_{oc} loss and must simultaneously provide high enough current to match the rear cell. Therefore, we also developed the wide bandgap material *m*-DTC-2F. Additionally, we conducted simulation on the optical bandgaps of rear and front cells and mainly discussed the rational in the selection of wide bandgap materials to match BTPV-4F on p. 7-8.

According to the reviewer's comments, we also add more discussion on the wide bandgap materials in the introduction in p. 5-6: "For the purpose of building highly efficient tandem OSCs, a wide bandgap sub-cell with suitable absorption range and low V_{oc} loss is also critical. Through the simulation of optical bandgaps of both cells, a front cell with wide bandgap of ca. 1.6 eV should be selected to ideally match the rear cell based on BTPV-4F. In this case, a new acceptor *m*-DTC-2F was developed with bandgap of 1.61 eV. The single-junction OSCs based on *m*-DTC-2F as the acceptor demonstrated a PCE of 12.2% with a high V_{oc} of 1.00 V and a J_{sc} of 17.1 mA cm⁻². Eventually, a high PCE of 16.40% was achieved for the tandem OSCs." More discussion on the development of the wide bandgap acceptor is added in p. 8: "Recently, Hsu *et al.* reported a carbazole based acceptor with two fluorine on the IC end groups called DTC(4Ph)-4FIC⁴⁰. The PM6: DTC(4Ph)-4FIC device shows high V_{oc} of 0.95 V and over 800 nm EQE response. In order to simultaneously get ideal bandgap and higher V_{oc} , we accordingly designed and synthesized a carbazole based small molecule acceptor *m*-DTC-2F with single fluoro-substitution IC end groups and *meta*-alkyl-phenyl side chains (Figure 1d)". In addition, discussion on the energy loss of tandem cells is added in p. 17: "In this work, *m*-DTC-2F has a V_{oc} loss of ca. 0.61 eV, which is one of the lowest values in the reported materials with similar optical bandgap of around 1.6 eV. However, in the current stage of tandem OSCs, higher percentage of V_{oc} loss still originates from the wide bandgap materials in front cells. Future works could focus more on the development of high performance wide bandgap materials to further boost the efficiency of tandem OSCs." and in p. 22: "So, in next step we should design

high performance front cell material systems with high FF (ca. 75%) in thick film device and low energy loss (ca. 0.55 eV)".

6. *The DFT calculation give HOMO and LUMO levels for the new BTPV-4F acceptor and Y6, but the values are never compared to the experimental data. That should have been done.*

Response: The mechanism of DFT calculation and cyclic voltammetry measurements have obvious difference. DFT calculation simulates the frontier orbital level of a single molecule. But for the cyclic voltammetry measurements, the energy level is estimated by measuring oxidation and reduction potential of solid films. However, the trends of energy levels of BTPV-4F and Y6 from the simulation and experiments match very well. We added the discussion of DFT calculation and experimental data in p. 9: "Both cases in DFT calculation and cyclic voltammetry measurements, BTPV-4F shows higher E_{HOMO} , similar E_{LUMO} and narrower bandgap than Y6. Such trend indicates that insertion of double bonds can effectively upshift the HOMO level and reduce the bandgap of acceptors."

7. *The authors should more clearly define the method on how they determine the optical band gap. Looking at the optical spectrum of the BTPV-4F film I would judge its onset at a higher photon energy than 1050 nm. This is also because some of the highlight of the paper is derived from Figure 2d, and a shift of the band gap to higher energy would result in a less remarkable result.*

Response: The optical bandgap of BTPV-4F was determined based on the position of onset of the UV-Vis absorption spectrum of BTPV-4F film. The onset was determined to be 1021 nm, corresponding to an optical bandgap of 1.21 eV just as claimed in the manuscript. This method on determination of optical bandgap was also adopted in all the reported works included in Figure 2d, which makes the comparison valid.

We thank the reviewer to point it out. We mentioned 1050 nm in the manuscript to indicate the whole coverage of the utilized absorption spectrum based on EQE, but the bandgap was not calculated based on 1050 nm. To remove this misunderstanding, we modified the sentences in the manuscript on p. 5 as : “BTPV-4F shows a significantly red-shifted absorption spectrum covering from 600 nm to 1050 nm. The optical bandgap of BTPV-4F was determined to be 1.21 eV based on the onset wavelength (1021 nm) of the UV-Vis absorption spectrum of BTPV-4F film”.

8. *Please also include in the text or Supp. Info the HOMO and LUMO levels of Y6, PM6, PTB7-Th, and PC71BM determined using the same methods, to give insight in the relative energy levels.*

Response: We added the cyclic voltammograms of these donor and acceptor material by using same methods, and their electronic energy levels measured by the electrochemical method in Supplementary Fig. 6. And we revised the sentence in p. 9 into: “In considering the complementary absorption and well-matched energy levels of the donor and acceptor materials (see Supplementary Fig. 6),” in the revised manuscript.

9. *Why is the V_{oc} enhanced using PC71BM?*

Response: In current stage of organic solar cells, the main energy loss comes from the large nonradiative recombination. It has been reported that, in many donor/acceptor system, nonradiative recombination could be reduced to a certain level by constructing ternary blend active layers. Especially for OSCs involving Y-series acceptor, it was widely confirmed that the energy loss can be effectively reduced with addition of the fullerene acceptor as the third component (*Adv. Mater.* **2019**, 31, 1902302). We also added discussion in the manuscript on p. 11: “The enhanced V_{oc} could be due to the reduced nonradiative recombination loss by

introducing PC₇₁BM, which can be widely observed in OSCs based on acceptors with the BTP central core^[47]».

10. Regarding the transient absorption I do not understand the lifetime of 60 ps. I judge it at about 20 ps. At the lifetime the intensity has decayed to 1/e ($e = \exp(1)$) and this happens at about 20 ps. How did the authors come to 60 ps? In addition, I would not consider 60 ps (and certainly not 20 ps) as a long lifetime. It is pretty short and will certainly be the limit exciton diffusion length. Some more discussion here would be useful.

Response: We added a sentence to explain the excited state lifetime of 60 ps in p.11: “(The decay data of the excited state were best fit to an $A \rightarrow B \rightarrow$ ground state (GS) kinetic model with time constants of $\tau_{A \rightarrow B} = 3.3 \pm 0.1$ ps and $\tau_{B \rightarrow GS} = 58.4 \pm 3.2$ ps (ca. 60 ps), so we think the lifetime is 60 ps)”. The excited state lifetime (~60 ps) of BTPV-4F is pretty long compared to the charge transfer lifetime (~2 ps) in the PTB7-Th:BTPV-4F blend film, and the calculated CT yield is $k_{CT}/(k_{CT} + k_{GS}) = 97\%$, which means the excitons dissociated sufficiently.

11. I do not see how a lower mobility can result in an increased fill factor (FF). The mobilities are quoted with three significant digits. Is that realistic? What are the error margins from the fits? Because the layer thickness goes with the power 3 in the SCLC equation an error (or variation) of 10 nm (see AFM data) (at 200 nm nominal thickness) already gives a 15% difference in the mobility. Also looking at the data in Supp. Fig. 10b the lower electron mobility of the blend with PC₇₁BM seems to be due to two or three data points at lower voltages. Furthermore I would like to see more evidence that a change in the mobility ratio of 1.48 to 1.06 has an effect on the FF. From the paper of Koster et al, in DOI: 10.1038/ncomms8083 it is not obvious that better balance of mobilities increase the FF. It could also be just a geometrical effect due to a higher V_{oc} . When a JV curve is shifted over the V-axis to higher V_{oc} , the FF will be higher.

Response: Thank the reviewer for pointing out the problem in measurement of mobility. The SCLC calculation is based on the equation of $J = 9\mu\epsilon_r\epsilon_0V^2/8d^3$ (where J is the current density, μ is the hole or electron mobility, V is the internal voltage in the device, ϵ_r is the relative dielectric constant of active layer material, ϵ_0 is the permittivity of empty space, and d is the thickness of the active layer). Among all these parameters, d contains least significant figures of two (for example, a film of around 120 nm should be as accurate as 0.00012 mm). Considering the rule of significant figures, when doing multiplication or division with measured values, the final result should have the same number of significant figures as the measured value with the least number of significant figures. In this case, μ should only contains significant figures of two, and we changed the values throughout the manuscript.

Throughout the fabrication process of hole/electron only devices for SCLC, random error always exists when repeating the experiment. To further evaluate the random error margin of the test, we adopted a pool of sample of 10 devices and the standard deviation of the mean was evaluated. The hole and electron mobilities of binary systems are estimated to be $(9.4\pm 0.6)\times 10^{-4} \text{ cm}^2\text{V}^{-1}\text{s}^{-1}$ and $(7.5\pm 0.7)\times 10^{-4} \text{ cm}^2\text{V}^{-1}\text{s}^{-1}$ and the best values are $1.1\times 10^{-3} \text{ cm}^2\text{V}^{-1}\text{s}^{-1}$ and $8.3\times 10^{-4} \text{ cm}^2\text{V}^{-1}\text{s}^{-1}$, respectively. As for the ternary system, the corresponding hole and electron mobilities are $(9.2\pm 0.4)\times 10^{-4} \text{ cm}^2\text{V}^{-1}\text{s}^{-1}$ and $(9.1\pm 0.6)\times 10^{-4} \text{ cm}^2\text{V}^{-1}\text{s}^{-1}$, and the best values are $1.0\times 10^{-3} \text{ cm}^2\text{V}^{-1}\text{s}^{-1}$ and $9.8\times 10^{-4} \text{ cm}^2\text{V}^{-1}\text{s}^{-1}$, respectively (the plots in Supplementary Fig. 10 showed the best measure mobility values among the 10 devices). In this case, we consider that electron mobilities increase obviously and hole mobilities show very slight decrease with the addition of PC₇₁BM. The new acquired statistical results shows the same trend as before. However, comparing with the previous data, the electron mobility has a larger degree of enhancement, while the hole mobility could be considered with negligible change (within the level of the standard deviation).

Based on the results of mobility, increased electron mobilities in ternary blends could be accountable for the higher FF in the ternary devices. In considering the reviewer's comments, we revised the sentence of "The more-balanced hole/electron mobilities should be accountable for

the higher FF in the ternary devices.” to “**The increased electron mobilities in ternary blends could be accountable for the higher FF in the ternary devices.**” (see p. 14)

12. In the in the in-plane and out-of-plane GIWAXS data one sees a clear (100) peak of PTB7-Th in the neat film at 0.6 \AA^{-1} , why is that absent in the blends? Likewise the blend has a signal at 0.8 \AA^{-1} that is not in the pristine material.

Response: The (100) peak of PTB7-Th is located at $q=0.3 \text{ \AA}^{-1}$ in the neat film and blend film in our work. The peak at 0.6 \AA^{-1} could be from higher order lamellar. The same level of ordering of the neat PTB7-Th cannot be expected in the blend film as the ordering could be affected from introducing a second material.

For the query related to the signal at 0.8 \AA^{-1} , we re-checked the data. The signals at 0.8 \AA^{-1} can be observed in the in-plane GIWAXS data in both pristine and blend films (Fig. 3(a)(b) and Supplementary Fig. 11(a)). The signal represents the polymer backbone.

13. Suppl. Table 8 does not provide the integrated intensities as suggested in the main text. Looking at Supp Fig. 12 I have hard time in seeing that the crystallinity increases. The text is also ambiguous there. It is stated that the higher crystallinity increases the mobility, but the authors actually showed earlier that the hole mobility decreases when P71BM is added. So I do not understand. The authors should also more clearly indicate which component we are main looking at in the GIWAXS. Is it PTB7-Th or BTPV-4F, or is it impossible to tell? This can be explained more clearly.

Response: The integrated intensities have provided in Supplementary Table 8. The increase of coherence length in ternary blends can prove the increased molecular ordering. The π - π coherence lengths (L) are estimated based on the Scherrer Equation ($L=2\pi K/\text{FWHM}$), where K is the shape factor (here we use 0.9), and FWHM is the full width at half maximum of the (010)

diffraction peaks. In Supplementary Fig. 12, the ternary film showed smaller full width at half maximum and increased CCL. However, the GIWAXS results of ternary films did not show obvious difference and just a slight increase of CCL, compared to binary films. In this case, the expression of “**improved molecular ordering**” in p. 15 may be more suitable. In the reviewer’s query 11, we re-measured the mobilities of binary and ternary systems. The results show that the electron mobility has a larger degree of enhancement, while the change of hole mobility could be considered within the level of the standard deviation, with the addition of PC₇₁BM. So we consider that the addition of PC₇₁BM could mainly influence the molecular packing of acceptors and further improve the CCL observed in blend films.

As for the component we are mainly looking at, the stacking peaks observed in both donor and acceptor neat films are around 1.7 Å⁻¹. The diffraction of donor and acceptor in the blend films were merged together and the stacking peak observed comes from combined PTB7-Th and BTPV-4F diffraction. So, it is hard to independently analyze the molecular packing of the donor PTB7-Th and the acceptor BTPV-4F in GIWAXS data of the blend films. From the results of mobilities, the improved molecular packing could be mostly attributed to the improved ordering of the acceptor rather than donor. We have added a note in the caption of Supplementary Table S8 that the (010) π - π stacking peaks of PTB7-Th and BTPV-4F are relatively close together and that the disorder is so large that the contributions in the blends cannot be separated..

14. In the discussion of the AFM the lower surface roughness with PC71BM is used as a reason for higher charge carrier generation due to better mixing. But there is no evidence for that. The current increase when PC71BM is due to absorption by PC71BM as can be seen in the EQE data of Figure 2c. In the other spectra range there is no increase in EQE and hence there is not better charge generation.

Response: We agree with the reviewer’s comment. The AFM's height image and roughness are not necessarily related to phase separation of donor and acceptor. It is too hard to claim “better

miscibility” merely based on the AFM's height image. To provide other evidence, we also measured the AFM phase image and TEM image, and the results were discussed in p. 15: “We also measured the AFM phase images and TEM images. As shown in Supplementary Fig. 13, binary and ternary blend films showed similar degree of phase segregation.”

However, smoother surface could favorably improve the contact between the interfacial layer and the active layer, leading to better performance of ternary OSCs. So we modified our expression in p. 15: “Smoother surface could favorably improve the contact between the interfacial layer and the active layer”.

15. Also the argument of an increase CCL being beneficial and better mixing being also beneficial seem at variance with each other. I know that all these arguments are frequently used in the literature, and mostly to show that they can be favorable. One could also state that if the lower surface roughness indicates better mixing the mobility would go down and that an increased CCL would limit charge generation. I think the authors should make a more balanced discussion.

Response: In binary and ternary systems, the CCL was characterized as 41 Å and 46 Å, respectively. Generally speaking, the difference is too small to influence the surface roughness. On the other hand, after re-evaluating the AFM phase and TEM images, we cannot safely claim the change of degree of the phase separation in binary and ternary systems, as we discussed in the reviewer’s Query 14. And we modified our expression in p. 15: “Smoother surface could favorably improve the contact between the interfacial layer and the active layer”.

16. What the absolute energy given the Fc/Fc+ couple to determine the HOMO and LUMO energies?

Response: We added a sentence to explain the calculation of the HOMO and LUMO energy

levels from the cyclic voltammograms in p. 9: “From the onset oxidation potential (E_{ox}) and onset reduction potential (E_{red}), HOMO energy level (E_{HOMO}) and the lowest unoccupied molecular orbital (LUMO) energy level (E_{LUMO}) were determined according to the equation $E_{\text{LUMO/HOMO}} = -e (E_{\text{red/ox}} + 4.36)$ (eV) where the unit of $E_{\text{red/ox}}$ is V vs. Ag/AgCl. (Redox potential of Fc/Fc^+ is 0.44 V vs Ag/AgCl in our measurement system (see Supplementary Fig. 6), and we take the energy level of Fc/Fc^+ as 4.8 eV below vacuum.)”

17. Why is surfactant added to PEDOT:PSS for spin coating on a MoO₃ layer? Which is that surfactant?

Response: The purpose of surfactant is to modify the wetting condition of the PEDOT:PSS solution and facilitate the spin-coating process on top of the MoO₃ surface. The surfactant is purchased from Zonyl FSO (a nonionic ethoxylated fluorosurfactant). We revised “surfactant” to “nonionic ethoxylated fluorosurfactant (Zonyl FSO)” in p. 26.

18. The device area is missing, so is the use of any masks. Please state if they are used or not and what is their relative to the nominal device area. How was the device area determined and how accurate?

Response: We added several sentences to describe the device area in p. 26: “The device area of the OSCs was 5.0 mm², which was defined by optical microscope (Olympus BX51). In order to accurately measure the photocurrent, a mask with an area of 4.80 mm² was used to define the effective area of the OSCs. The devices with or without mask showed consistent photovoltaic performance values with relative errors within 0.3%.”

19. In the end I would like to come back to the discussion on which cell is now limiting more. The wide band gap cell has a voltage loss of 1.6-1.0 = 0.60 eV, the narrow band gap cell has a

voltage loss of $1.20 - 0.67 = 0.53$ eV. I think that at present the best OSCs have larger Voc losses for higher band gaps. This would actually argue that it is more important to develop wide bandgap sub cells than narrow band gap sub cells. A more detailed discussion on that could make this paper much more solid and important.

Response: We appreciate the review's comments and suggestion, and we agree that the wide bandgap materials are also very important to tandem OSCs. As shown in our response to Query 5, we added more discussions on this issue.

Our logic flow in constructing the tandem OSC in this work was based on the understanding of Shockley-Queisser limit in solar cells. We firstly focused on the development of narrow bandgap acceptor to purposely maximize the utilization of solar spectrum. In a certain combination of front and rear cell materials, narrow bandgap material usually affects the ultimate overall absorbing potential by determining the absorbing range. Subsequently, lowering the energy loss of those photons absorbed should be considered (especially for high energy photons), which is certainly more determined by wide bandgap cell. As discussed in the manuscript, future works should focus more on the development of wide bandgap materials to further boost the efficiency of tandem OSCs.

Response to Reviewer #3:

This paper developed a new structured non-fullerene acceptor material that can absorb the IR region and applied it to organic solar cells. The development of high-performance photoactive materials capable of absorbing light in various wavelength regions is an important means of ultimately increasing the usefulness of organic solar cells, colorfully and transparently, so it can be said to be a meaningful study. Therefore, although the logic of introducing new materials used the existing method and the tandem device did not show the best performance at the present time, it is judged that there is enough novelty to be accepted on Nature Commun. However, the

followings should be corrected and supplemented.

On page 11, the bleach peak at 713 nm in Sup Fig. 7(c) is claimed to be matched with the absorption peak of PTB7-Th. In order to explain this logic, in addition to the movement of holes from excited BTPV-4F to PTB7-Th's HOMO, explanation on how electrons can be introduced into PTB7-Th's LUMO should be added.

Response: Thanks for the reviewer's comments and suggestion. In the fs-TA experiment, the excitation wavelength is 850 nm and will selectively excite the acceptor BTPV-4F in the blend, as shown in Scheme R1 below. PTB7-Th is not excited so the electrons will not be introduced to its LUMO. The bleach peak of PTB7-Th comes from the hole transfer from BTPV-4F excited state to PTB7-Th ground state, generating (BTPV-4F)⁻ and (PTB7-Th)⁺ charge transfer state. The formation of (PTB7-Th)⁺ reduced the population of PTB7-Th ground state, resulting in the bleach signal in the transient absorption spectra.

Schemem R1: Diagram summarizing the possible electron transfer processes.

On page 14, the phase segregation of donor and acceptor was inferred from the AFM's Topo image, but it was insufficient to confirm the phase segregation from the Topo image. To be more accurate, additional data of AFM phase image and TEM image are required.

Response: We added AFM phase images and TEM images and modified our expression in p.15: "Smoother surface could favorably improve the contact between the interfacial layer and the

active layer. We also measured the AFM phase images and TEM images. As shown in Supplementary Fig. 13, binary and ternary blend films showed similar degree of phase segregation.”

On page 13, it was explained that J_{ph}/J_{sat} indicates charge generation and exciton dissociation efficiency in Sup. Fig. 9, but it seems reasonable that the expression of charge collection is also included.

Response: We added discussion of charge collection in p. 14: “As shown in Supplementary Fig. 9, under the short-circuit condition, the J_{ph}/J_{sat} ratios are 0.982 for the binary device and 0.986 for the ternary device; under maximal power output conditions, the ratios are 0.839 and 0.858, respectively. The results indicate the ternary devices have the higher both exciton dissociation and charge collection efficiency.”

On page 10, the number of significant digits in V_{oc} varies from data to data (0.65 V and 0.671 V), please use the same significant digit.

Response: The data have been modified in same significant digits in the revised manuscript.

REVIEWERS' COMMENTS

Reviewer #1 (Remarks to the Author):

Authors have well addressed reviewer comments. The acceptance of this MS for publication can be recommended. Note that a survey of narrow bandgap acceptors has been discussed in maintext and SI, wherein some of highly relevant reference is missed and can be incorporated, such as NIR acceptor with 1.23 eV of optical gap and 13.75% PCE in single junction OSCs, <https://doi.org/10.1039/D0TA06907H>.

Reviewer #3 (Remarks to the Author):

Except for the femtosecond transient absorption spectra, all questions have been resolved. Your attached Schemem R1 is confirmed. However, I don't think the bleach peak can appear with PTB7-Th + itself. So, I think electrons must be introduced into the LUMO of PTB7-Th+ in order for the bleach peak to appear. How can electrons can be introduced into the LUMO of PTB7-Th+? It seems strange to me that a large bleaching peak appears in the 737nm region, which is a higher energy region than the excited 850nm. If there was references for such an example, please refer to it.

Response letter to reviewers:

Response to Reviewer #1:

Authors have well addressed reviewer comments. The acceptance of this MS for publication can be recommended. Note that a survey of narrow bandgap acceptors has been discussed in maintext and SI, wherein some of highly relevant reference is missed and can be incorporated, such as NIR acceptor with 1.23 eV of optical gap and 13.75% PCE in single junction OSCs, <https://doi.org/10.1039/D0TA06907H>.

Response: We appreciate the review's comments and suggestions. The recommended reference paper was included in the survey of narrow bandgap acceptors in **Figure 4** and **Supplementary**

Table 7 and cited in Ref. 15 in Supplementary information.

Response to Reviewer #3:

Except for the femtosecond transient absorption spectra, all questions have been resolved.

Your attached Scheme R1 is confirmed. However, I don't think the bleach peak can appear with PTB7-Th + itself. So, I think electrons must be introduced into the LUMO of PTB7-Th⁺ in order for the bleach peak to appear. How can electrons can be introduced into the LUMO of PTB7-Th⁺?

It seems strange to me that a large bleaching peak appears in the 737nm region, which is a higher energy region than the excited 850nm. If there was references for such an example, please refer to it.

Response: Thank the reviewer for the question. The pump wavelength was set to 850 nm in order to avoid co-excitation of PTB7-Th. Moreover, since the singlet energy of BTPV-4F is lower than that of PTB7-Th, no energy transfer could happen from BTPV-4F excited state to PTB7-Th. Therefore the PTB7-Th excited state, in which the electron is introduced into its LUMO, would not form in our fs-TA experiment. We believe the best explanation for the bleach peak at 737 nm is the formation of (PTB7-Th)⁺ resulted from ultrafast hole transfer. The bleach peak of PTB7-Th comes from the hole transfer from BTPV-4F excited state to PTB7-Th ground state, generating (BTPV-4F)⁻ and (PTB7-Th)⁺ charge transfer state.

Observation of donor bleaching peak right after excitation of acceptor at lower energy was widely observed in fs-TA experiments. Similar results for other polymer donors were also reported: for example, J71 (*Nat. Commun.* **2016**, 7, 13651), PTQ10 (*J. Am. Chem. Soc.* **2020**, 142, 1465) and PM6 (*Adv. Energy Mater.* **2019**, 9, 1901728). Peak shape, intensity and kinetics may vary due to different donor-acceptor systems.